# FLOW ACTOR-CRITIC FOR OFFLINE REINFORCEMENT LEARNING

**Jongseong Chae[1]**  **Jongeui Park[1]**  **Yongjae Shin[1]**  **Gyeongmin Kim[1]**
**Seungyul Han[2]**  **Youngchul Sung[1,*]**
[1]KAIST [2]UNIST
{jongseong.chae,ycsung}@kaist.ac.kr

## ABSTRACT

The dataset distributions in offline reinforcement learning (RL) often exhibit complex and multi-modal distributions, necessitating expressive policies to capture such distributions beyond widely-used Gaussian policies. To handle such complex and multi-modal datasets, in this paper, we propose Flow Actor-Critic, a new actor-critic method for offline RL, based on recent flow policies. The proposed method not only uses the flow model for actor as in previous flow policies but also exploits the expressive flow model for conservative critic acquisition to prevent Q-value explosion in out-of-data regions. To this end, we propose a new form of critic regularizer based on the flow behavior proxy model obtained as a byproduct of flow-based actor design. Leveraging the flow model in this joint way, we achieve new state-of-the-art performance for test datasets of offline RL including the D4RL and recent OGBench benchmarks.

## 1 INTRODUCTION

Offline Reinforcement Learning (RL) seeks an optimal policy from a pre-collected dataset without additional environment interactions, which may be costly or unsafe in real-world situations (Lange et al., 2012; Levine et al., 2020). Despite such advantages, offline RL suffers from value overestimation for out-of-distribution (OOD) actions due to its limited dataset coverage, which degrades performance. Many methods have been proposed to deal with this OOD value overestimation based on techniques such as constraining the target policy near the behavior policy of the dataset (Kumar et al., 2019; Fujimoto et al., 2019; Wu et al., 2022; Fujimoto & Gu, 2021; Nair et al., 2020) or suppressing the value function in the OOD region with penalization (Kumar et al., 2020; Mao et al., 2023). These methods have been shown to be effective for many offline RL tasks.

As offline RL datasets accumulate and grow diverse, however, their behavioral distributions become complicated and sometimes multi-modal. Thus, simple behavior policy modeling becomes insufficient (Park et al., 2024b) and more expressive policies are required to model the behavior policy and its support. For example, diffusion policies adopted from the vision domain have garnered strong attention from the RL community as a candidate for expressive policies (Wang et al., 2022; Chen et al., 2022; Hansen-Estruch et al., 2023; Chen et al., 2024; Zhang et al., 2025). Diffusion policies learn the underlying data distribution using forward and reverse processes and can be trained efficiently by score matching with noise models (Ho et al., 2020; Song et al., 2020). They have shown impressive offline performance. Even with this advantage, diffusion models have a limitation that they can be used to construct an actor capable of sampling actions close to the behavior policy, but the iterative nature of their multi-step processing makes policy optimization unstable (Park et al., 2025)

Therefore, Park et al. (2025) recently adopted the *flow model* (Dinh et al., 2014; Rezende et al., 2014; Dinh et al., 2016) as an alternative for expressive policies. Although their work proposed one-step flow actor and demonstrated promising results, they did not exploit the full advantage of the flow policy. They only considered the actor design to maximize the conventional Q function under a constraint on the distance of the target policy from a flow-based behavior policy estimate, but overlooked the key fact that an accurate behavior density estimate itself is available with the flow model in contrast to diffusion models.

---

*Corresponding author. Code available at https://github.com/JongseongChae/FAC.

In this paper, we adopt the flow model and fully exploit it in a joint way to optimize an offline RL policy. We not only use the flow behavior policy model as the distance anchor of the target policy but also use it for Q function penalization in the OOD region by identifying the OOD region based on the flow behavior density, which provides strong density contrast to distinguish ID region from OOD one. Note that it has long been considered in offline RL that identifying the OOD region is difficult, and many previous works circumvent this difficulty with indirect methods (Kumar et al., 2020; Mao et al., 2023; Wu et al., 2022). However, the highly expressive flow behavior model and accompanying density provide us with a rare opportunity to directly tackle this problem. Our flow actor-critic (FAC) designed in such a way exhibits outstanding performance in many current offline benchmark tasks.

## 2 PRELIMINARIES

### 2.1 OFFLINE REINFORCEMENT LEARNING

An RL problem is formulated as a Markov Decision Process (MDP) $\mathcal{M} = (\mathcal{S}, \mathcal{A}, P, \rho_0, r, \gamma)$, where $\mathcal{S}$ is the state space, $\mathcal{A}$ is the $d_{\mathcal{A}}$-dimensional action space, $P(s'|s, a)$ is the transition dynamics, $\rho_0$ is the initial state distribution, $r(s, a)$ is the reward function, and $\gamma$ is the discount factor (Sutton & Barto, 2018). The goal of offline RL is to find a policy $\pi : \mathcal{S} \rightarrow \Delta(\mathcal{A})$ that maximizes the expected discounted return $J(\pi_\theta) = \mathbb{E}_{\rho_0, P, \pi} \left[ \sum_{t=0}^{\infty} \gamma^t r(s_t, a_t) \right]$ from a static dataset $\mathcal{D} = \{(s, a, r, s')\}$ collected by an unknown underlying behavior policy $\beta$. For a policy $\pi$, the action value function is defined as $Q^\pi(s, a) = \mathbb{E}_\pi \left[ \sum_{t=0}^{\infty} \gamma^t r(s_t, a_t) | s_0 = s, a_0 = a \right]$ and can be estimated by iterating the Bellman operator $\mathcal{T}^\pi Q(s, a) = r(s, a) + \gamma \mathbb{E}_{s' \sim P(\cdot|s,a), a' \sim \pi(\cdot|s')} [Q(s', a')]$.

**Support-Constrained Policy Optimization.** The main challenge of offline RL is $Q$-value overestimation in the OOD action region outside the support of $\beta$ (Kumar et al., 2020; Mao et al., 2023). A typical approach to this problem is policy optimization under a distance constraint between the target policy and the behavior policy (Fujimoto et al., 2019; Peng et al., 2019; Wu et al., 2019; 2022):

$$\max_\pi \ \mathbb{E}_{s \sim \mathcal{D}, a \sim \pi} [Q(s, a)] \quad \text{s.t.} \quad \mathbb{E}_{s \sim \mathcal{D}} [D(\pi(\cdot|s), \beta(\cdot|s))] < \delta, \tag{1}$$

where $D$ is some distance or divergence measure. The rationale behind eq. (1) is that by restricting the distance of $\pi$ from $\beta$, the action from $\pi$ does not fall into the OOD region of the estimated $Q$, avoiding the overestimated region of $Q$. Practically, the hard constraint in eq. (1) is replaced with a regularization term, yielding the following optimization (Kumar et al., 2019; Fujimoto & Gu, 2021):

$$\max_\pi \ \mathbb{E}_{s \sim \mathcal{D}, a \sim \pi} [Q(s, a)] - \lambda \, \mathbb{E}_{s \sim \mathcal{D}} [D(\pi(\cdot|s), \beta(\cdot|s))], \tag{2}$$

where $\lambda > 0$ is a regularization coefficient. Note that eq. (2) can be viewed as the Lagrangian of the problem (1).

### 2.2 FLOW MODELS AND FLOW-BASED POLICIES

Normalizing flows are a popular framework for modeling complex data distributions (Dinh et al., 2014; Rezende et al., 2014). A continuous normalizing flow (CNF) is defined as a bijective function $\phi_u : \mathbb{R}^d \rightarrow \mathbb{R}^d$ (Chen et al., 2018), specified by a time-dependent vector field $v_u$ via the 1st-order ordinary differential equation:

$$\frac{d}{du} \phi_u(x) = v_u(\phi_u(x)), \quad \phi_0(x) = x, \tag{3}$$

yielding

$$x_u = \phi_u(x_0) = x_0 + \int_0^u v_t(x_t) dt. \tag{4}$$

Given data samples $x_1 \sim p_1(x)$ and a simple base distribution $p_0$ (e.g., standard normal), we want to learn the mapping $\phi_u$ such that $\phi_0(x_0) = x_0$ at time $u = 0$ is transported to $\phi_1(x_0) = x_1$ at time $u = 1$. Since $\phi_1(\cdot)$ is a deterministic function, the density of $p_1$ can easily be computed by the change of variables: $\log p_1(x_1) = \log p_0(x_0) - \log \left| \det \left( \frac{\partial \phi(x_0)}{\partial x_0} \right) \right|$. It is shown that the determinant of the Jacobian can be replaced with the divergence of the vector field $v_u$ (Chen et al., 2018; Ben-Hamu et al., 2022). Thus, the data distribution density can be written as $\log p_1(x_1) = \log p_0(x_0) - \int_0^1 \nabla_{x_u} \cdot v_u(x_u) du$.

A recent efficient way to learn a flow for a set of pairs $\{(x_0, x_1)\}$ is flow matching (FM), not requiring complicated likelihood maximization (Lipman et al., 2022). FM learns the velocity field such that for a pair $(x_0, x_1)$, the instantaneous movement velocity vector $v_u(x_u)$ matches the overall displacement vector $x_1 - x_0$ along the points on the line connecting $x_0$ and $x_1$, i.e., $x_u = (1 - u)x_0 + ux_1$. Thus, when the velocity vector field is parameterized with $\theta$, the loss for learning $\theta$ is given by

$$\min_\theta \ \mathbb{E}_{u \sim \text{Unif}([0,1]), \ x_1 \sim p_1(x), \ x_0 \sim p_0(x)} \left[ \|v_\theta(x_u; u) - (x_1 - x_0)\|_2^2 \right], \tag{5}$$

Once $v_\theta$ is trained, samples from $p_1$ can be generated by solving eq. (4) with an integral solver by setting $u = 1$, and the density of the samples is available as aforementioned.

**Flow-based Policy.** Constructing a policy from the flow model is straightforward. One can define the flow function from the action space to the action space, depending on both time $u$ and state $s$. Then, by pairing $z \sim \mathcal{N}(0, I)$ and action $a$ (from the behavior policy), the corresponding velocity vector field parameterized with $\psi$ is learned with the following loss (Lipman et al., 2022; Park et al., 2025):

$$\min_\psi \ \mathbb{E}_{(s,a) \sim \mathcal{D}, z \sim \mathcal{N}(0,I), u \sim \text{Unif}([0,1])} \left[ \|v_\psi(\tilde{a}_u; s, u) - (a - z)\|_2^2 \right], \tag{6}$$

where $\tilde{a}_u = (1 - u)z + ua$. We refer to the so-learned flow policy with $(s, a) \sim \mathcal{D}$ as the *behavior proxy policy* $\hat{\beta}_\psi(\cdot|s)$. Then, the behavior proxy density is given by

$$\log \hat{\beta}_\psi(a|s) = \log p_0(\hat{z}) - \int_0^1 \nabla_{a_u} \cdot v_\psi(a_u; s, u) du. \tag{7}$$

One thing to note is that samples from a flow policy are stochastic samples for given $s$ due to the initial condition $z \sim \mathcal{N}(0, I)$ even though the flow function itself is deterministic. Thus, we will use the notation $a(s, z)$ to show this dependency in the case of flow action, if necessary.

## 3  MOTIVATION: STRENGTH OF FLOW BEHAVIOR PROXY DENSITY

Now let us investigate how well the flow behavior proxy policy $\hat{\beta}_\psi(\cdot|s)$ tracks the actual behavior policy $\beta$ in terms of both sampling and density estimation. For this, we conducted an experiment with $\beta$ being a 2-D Gaussian mixture with four modes, as shown in the leftmost plot in Fig. 1. For comparison, we considered four other behavior cloning (BC) models: simple Gaussian, conditional VAE (Kingma & Welling, 2013), and diffusion models with 10 and 50 denoising steps (Ho et al., 2020; Sohl-Dickstein et al., 2015; Wang et al., 2022). For the flow model, we used the Euler method with 10 steps as an integral solver for both sampling and density evaluation (Lipman et al., 2022; Chen et al., 2018). The results are shown in Fig. 1. In the case of VAE and diffusion models, we show the ELBO since the density is not directly available. Details of the experiment are in Appendix C.

As expected, simple Gaussian with single mode cannot distinguish the in-distribution (ID) region with its peak at the center of the four true modes on which the actual density is almost zero. VAE finds the four modes but generates noticeable samples outside the dataset support. Furthermore, amortized inference places probability density in low-density areas when the latent variable does not

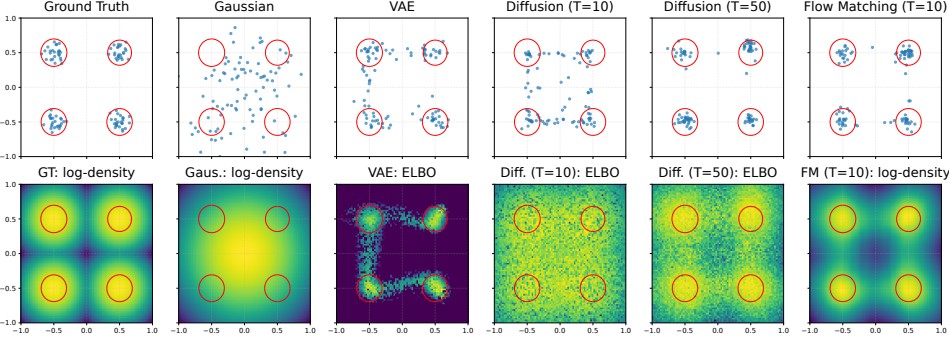

Figure 1: BC models on a synthetic four-component Gaussian mixture dataset: Top row - samples from each BC model. Bottom row - log-density or ELBO plot.

match the true posterior distribution, and hence the obtained ELBO is quite different from the true density as seen in the bottom row of the figure. The diffusion model with 50 denoising steps tends to generate samples within the in-distribution region, while the performance degrades with 10 denoising steps. Note that the obtained ELBO of density is bad, almost not separating the four modes to make it not suited for a density estimator, whereas the sampling performance is satisfactory. In contrast, the flow behavior proxy model shows good sampling performance and, importantly, yields a very strong estimate of the true density. Thus, the flow model can be used not only for an actor but also as a density estimator that can be used to distinguish the ID and OOD regions. In the following section, we present our algorithm to design both actor and critic, exploiting these two facts.

# 4 METHOD: FLOW ACTOR-CRITIC

## 4.1 CRITIC PENALIZATION WITH FLOW BEHAVIOR PROXY DENSITY

A typical way to prevent overestimation of Q values in the OOD region for offline RL is to learn a penalized Q function. CQL (Kumar et al., 2020) and SVR (Mao et al., 2023), two representative methods, use the following losses:

$$\min_Q \ \alpha \left(\mathbb{E}_{s\sim\mathcal{D},a\sim\pi}\left[Q(s,a)\right] - \mathbb{E}_{s\sim\mathcal{D},a\sim\beta}\left[Q(s,a)\right]\right) + \text{TD Loss}$$

and

$$\min_Q \alpha \left(\mathbb{E}_{s\sim\mathcal{D},a\sim\zeta}[(Q(s,a) - Q_{min})^2] - \mathbb{E}_{s\sim\mathcal{D},a\sim\beta}\left[\frac{\zeta(a|s)}{\beta(a|s)}(Q(s,a) - Q_{min})^2\right]\right) + \text{TD Loss},$$

respectively, where the TD loss is given by $\mathbb{E}_{s,a,s'\sim\mathcal{D}}\left[(Q(s,a) - \mathcal{T}^\pi Q(s,a))^2\right]$, and $\zeta(a|s)$ in SVR is a distribution covering the entire action range. It is known that the CQL penalization does not properly implement the correct Bellman operator in the ID region (Mao et al., 2023). SVR fixes this problem by using the difference of importance sampling (IS) integration when $\beta$ does not dominate $\zeta$. Although SVR circumvents direct OOD region identification intelligently using the IS technique, the main drawback is simple modeling of $\beta$, such as Gaussian density, required to compute the IS ratio. As seen in the previous section, when the true behavior distribution is complicated and multi-modal, Gaussian modeling places high non-zero density on regions with nearly zero actual density. In such cases, the IS ratio blows up incorrectly, deteriorating the performance, as shown in Appendix D.2.

With the flow behavior proxy density $\hat{\beta}_\psi$, a strong estimator for the true behavior density $\beta$ at our hands, we can directly identify the ID and OOD regions and control the Q penalization term. To this end, we define the weight $w^{\hat{\beta}_\psi}(s,a)$ as

$$w^{\hat{\beta}_\psi}(s,a) = \max(0, \ 1 - (\hat{\beta}_\psi(a|s)/\epsilon)) \quad \text{for some } \epsilon > 0. \quad (8)$$

Figure 2: weight $w^{\hat{\beta}}$

The weight $w^{\hat{\beta}_\psi}(s,a)$ vanishes in the well-supported region with $\hat{\beta}_\psi(a|s) \geq \epsilon$, and increases linearly as $\hat{\beta}_\psi$ decreases below the threshold $\epsilon$ towards zero. The threshold $\epsilon$ is introduced because $\hat{\beta}_\psi$ is not perfect even though it is strong. We want to exclude the weak spurious density from the ID region but gradually. With this weight, we propose the following loss for critic learning:

$$\min_Q \ \alpha \, \mathbb{E}_{s\sim\mathcal{D}, \ a\sim\pi}\left[w^{\hat{\beta}_\psi}(s,a)Q(s,a)\right] + \text{TD Loss}, \quad (9)$$

where $\alpha > 0$ controls the strength of penalization. In eq. (9), our penalization term is not in the form of some difference as in CQL or SVR, but directly pinpoints the OOD region based on the strong behavior density estimator $\hat{\beta}_\psi$. When $\hat{\beta}_\psi(a|s) \geq \epsilon$, i.e., $a \in \text{support}(\beta)$ with confidence, the penalization term disappears. However, the penalization term gradually kicks in as the confidence decreases below $\epsilon$.

**Proposition 1.** *Let $\beta$ be the underlying behavior policy, $\hat{\beta}$ be our proxy for $\beta$, $\pi$ be the learned actor, and $Q$ be the value function of $\pi$. Consider the original Bellman operator $\mathcal{T}^\pi Q(s,a) = r(s,a) + \gamma\mathbb{E}_{s'\sim P(\cdot|s,a), \ a'\sim\pi(\cdot|s')}\left[Q(s',a')\right]$. Then, in the tabular setting without function approximation, the*

*objective (9) yields the following operator:*

$$
\mathcal{T}_{FAC}^\pi Q(s,a) = \begin{cases} \mathcal{T}^\pi Q(s,a) & \text{if } \hat{\beta}(a|s) \geq \epsilon, \ \beta(a|s) > 0 \\ \mathcal{T}^\pi Q(s,a) - \frac{\alpha}{2} \left( \frac{w^{\hat{\beta}}(s,a)\pi(a|s)}{\beta(a|s)} \right) & \text{if } \hat{\beta}(a|s) < \epsilon, \ \beta(a|s) > 0 \\ -\infty & \text{if } \beta(a|s) = 0. \end{cases} \tag{10}
$$

*unless $\beta(a|s) = 0$ and $w^{\hat{\beta}}(s,a) = 0$ simultaneously.*

By the definition of the weight $w^{\hat{\beta}_\psi}(s,a)$, it is highly unlikely that we have $\beta(a|s) = 0$ and $w^{\hat{\beta}}(s,a) = 0$ simultaneously. The operator associated with our new penalization maintains the original Bellman operator inside most of the ID support region, while strongly suppressing the Q value in the OOD region. Furthermore, for a weak confidence of the estimator, it gradually suppresses the Q value according to its confidence. Thus, when the proxy $\hat{\beta}_\psi$ is a strong estimator of $\beta$, the proposed penalization works properly. The proof of Proposition 1 and a rigorous analysis of the operator $\mathcal{T}_{FAC}^\pi$ is provided in Appendix E, showing that the operator $\mathcal{T}_{FAC}^\pi$ has a unique fixed point, yielding unbiased Q values for state-action pairs with $\hat{\beta}_\psi(a|s) \geq \epsilon$ and underestimated conservative Q values for those with $\hat{\beta}_\psi(a|s) < \epsilon$.

Note that in our critic penalization, determining $\epsilon$ is important. We consider two dataset-driven methods for $\epsilon$ design: (1) dataset-wide constant threshold $\min_{(s,a)\sim\mathcal{D}} \hat{\beta}_\psi(a|s)$ and (2) batch-adaptive threshold $\hat{\beta}_\psi(a|s)$ using mini-batch samples $\mathcal{B} = \{(s,a)\}$ from $\mathcal{D}$. The dataset-wide threshold does not exclude some of the actual ID region of the dataset, being suited to datasets with wide state coverage and low action diversity, and the batch-adaptive threshold adapts to local coverage and multi-modality, being suited to datasets with limited state coverage and high action diversity.

## 4.2 ENHANCED FLOW ACTOR OPTIMIZATION

The typical offline RL actor optimization is done with the objective (2). That is, the objective is to maximize Q value while keeping near the behavior policy $\beta$. One can apply this offline actor design principle to flow-based actor (Park et al., 2025). In this case, the flow behavior proxy policy $\hat{\beta}_\psi$ is learned as described in Section 2.2. Then, the objective (2) for the actual target policy $\pi_\theta$ can be implemented as follows (Park et al., 2025):

$$
\max_{\pi_\theta} \mathbb{E}_{s\sim\mathcal{D}, a\sim\pi_\theta} \left[ Q_\phi(s,a) \right] - \lambda \mathbb{E}_{s\sim\mathcal{D}, z\sim\mathcal{N}(0,I)} \left[ \|a_\theta(s,z) - a_\psi(s,z)\|_2^2 \right], \tag{11}
$$

where $a_\theta(s,z)$ and $a_\psi(s,z)$ denote actions generated by $\pi_\theta$ and $\hat{\beta}_\psi$ given $(s,z)$, respectively. Here, $a_\psi(a,z)$ is realized with velocity modeling (6) and integration (4) for full expressibility. However, the target policy $\pi_\theta$ implementation is simplified because the numerical multi-step integration based on velocity modeling makes gradient backpropagation complicated. Thus, Park et al. (2025) used a one-step flow policy, which transports $z$ directly to $a$ via a flow $a_\theta(\cdot\,; s, u=1): \mathcal{A} \to \mathcal{A}: z \mapsto a$. Although this one-step flow policy $\pi_\theta$ has less expressibility than the multistep behavior proxy policy $\hat{\beta}_\psi$, it still has sufficient capability to express multi-modality, as shown in Appendix D.2. Thus, we adopt this one-step flow policy as our actor too.

In most cases of policy optimization via Q maximization with a distance constraint like (11) including Park et al. (2025), simple $Q_\phi(s,a)$, just trained with the Bellman error $\mathbb{E}_{s,a,s'\sim\mathcal{D}} \left[ (Q_\phi(s,a) - \mathcal{T}^{\pi_\theta} Q_{\bar{\phi}}(s,a))^2 \right]$, is used, where $\bar{\phi}$ is the target Q network parameter. The rationale behind this is that the distance regularization term in (11) attracts the policy to the behavior support although the Q function is overestimated in the OOD region. However, when near-optimal actions are rare compared to sub-optimal ones in the dataset, there is a problem. Weak regularization in (11) drives value maximization toward the OOD region on which unpenalized $Q_\phi$ has overestimated values, yielding wrong actions. On the other hand, strong regularization induces the actor to imitate sub-optimal actions mostly. In such cases, if we use weak distance regularization and a Q function well-penalized at the OOD region in (11), then maximizing such a Q function yields good actions within the ID support. For such an idea to work, we need a Q function estimate precisely penalized at the OOD region. Therefore, we use the critic penalized with highly expressive flow behavior proxy proposed in Section 4.1. We name this new actor-critic structure for offline RL based on the flow model *Flow Actor-Critic (FAC)*. It will be shown that FAC sets new state-of-the-art performance for difficult tasks of the OGBench in the experiment section soon.

### 4.3 PRACTICAL IMPLEMENTATION

The practical implementation of FAC is provided in Algorithm 1. The proposed method comprises three key components: a flow behavior proxy policy, a one-step flow actor, and twin critics to ensure stable $Q$ value estimation (Fujimoto et al., 2018). The training procedure consists of two stages.

In the first stage, we train a flow behavior proxy policy $\hat{\beta}_\psi$ by eq. (6). After training the proxy policy, we evaluate behavior proxy density by eq. (7) for all state-action pairs in the dataset and store these evaluations along with the transitions. These stored evaluations as the dataset-driven threshold allow efficient identification of ID and OOD regions in the subsequent training stage.

In the second stage, we train penalized critics and a one-step flow actor concurrently. The critics $Q_\phi$ are trained by eq. (9) with target critics for stable learning. Specifically, the proposed critic penalization applies behavior-aware weights based on the behavior proxy density of actor actions and the stored evaluations of the dataset. The actor $\pi_\theta$ is trained by eq. (11).

---

**Algorithm 1:** Flow Actor-Critic

Initialize one-step flow policy network $\pi_\theta$, critic networks $Q_{\phi_1}, Q_{\phi_2}$, target networks $Q_{\bar{\phi}_1}, Q_{\bar{\phi}_2}$, and flow behavior proxy network $\hat{\beta}_\psi$

**for** *each iteration* **do**
    Sample $(s, a) \sim \mathcal{D}$, $z \sim \mathcal{N}(0, I)$, $u \sim \text{Unif}([0, 1])$.
    Update $\hat{\beta}_\psi$ with eq. (6).

Compute $\{\hat{\beta}_\psi(a|s)\}$ for the samples in $\mathcal{D}$ by eq. (7) and store the evaluations to $\mathcal{D}$.

**for** *each iteration* **do**
    Sample $(s, a, r, s') \sim \mathcal{D}$, $z \sim \mathcal{N}(0, I)$ and compute $\epsilon$.
    Update $Q_{\phi_i}$ with eq. (9), $\quad i \in \{1, 2\}$.
    Update $\pi_\theta$ with eq. (11).
    $\bar{\phi}_i \leftarrow \rho\phi_i + (1 - \rho)\bar{\phi}_i, \quad i \in \{1, 2\}$ for some $\rho$.

---

## 5 RELATED WORKS

**Offline RL with Gaussian policies.** Most methods with Gaussian policies can be grouped into two complementary approaches: actor regularization (AR) and critic penalization (CP). AR methods constrain the learned policy to the behavior policy in the dataset, such as TD3+BC (Fujimoto & Gu, 2021) imposing a simple behavior cloning regularizer and SPOT (Wu et al., 2022) using VAE-based ELBO to induce high-support actions. Other related AR methods extract the policy through value-weighted regression (Peng et al., 2019; Nair et al., 2020; Lee et al., 2021), such as IQL (Kostrikov et al., 2021). On the other hand, CP methods, such as CQL (Kumar et al., 2020), penalize value estimates for policy actions, but the induced policy becomes overly conservative. To mitigate this, MCQ (Lyu et al., 2022) introduces pseudo target value, EPQ (Yeom et al., 2024) uses state-dependent penalty weights defined by VAE-based ELBO and a current policy, SAC-RND (Nikulin et al., 2023) leverages RND-based anti-exploration bonus (Burda et al., 2018), and SVR (Mao et al., 2023) uses the property of importance sampling. Based on these approaches, ReBRAC (Tarasov et al., 2023) shows that complementary use of AR and CP (Wu et al., 2019; Kim et al., 2025) yields strong performance when this joint approach is integrated with modern offline RL design choices.

**Diffusion and flow-based policies.** Expressive policies have recently emerged as alternatives for modeling complex action distributions. The methods using such expressive policies can be categorized by whether they maximize their $Q$ function explicitly or implicitly. For diffusion-based methods, explicit approaches, such as CAC (Ding & Jin, 2024) and SRPO (Chen et al., 2024), optimize $Q$-maximizing policies (Wang et al., 2022; He et al., 2023; Zhang et al., 2024). Implicit approaches, such as IDQL (Hansen-Estruch et al., 2023), either use value-weighted regression (Lu et al., 2023; Kang et al., 2023; Ding et al., 2024) or apply value-weighted sampling (Chen et al., 2022; He et al., 2024). For flow-based methods, FQL (Park et al., 2025) trains a $Q$-maximizing policy using the distance regularizer with a flow behavior proxy policy. Furthermore, Park et al. (2025) has shown that stable policy extraction for multistep flow policies is challenging: value-weighted regression (FAWAC), explicit $Q$-maximization (FBRAC), and value-weighted sampling (IFQL). Another method, QIPO-OT (Zhang et al., 2025), trains a flow matching policy through value-weighted regression.

Table 1: Evaluation over 55 singletasks of the OGBench. For each task category, we report the final performance averaged across its 5 singletasks, over 8 seeds (4 seeds for pixel-based tasks), with $\pm$ indicating the standard deviation. Full evaluation results on the 55 singletasks are in Table 3.

| Task Category | Gaussian Policies | | | Diffusion Policies | | | Flow Policies | | | | |
|---|---|---|---|---|---|---|---|---|---|---|---|
| | BC | IQL | ReBRAC | IDQL | SRPO | CAC | FAWAC | FBRAC | IFQL | FQL | FAC (Ours) |
| antmaze-large-navigate | 10.6 | 53.4 | 80.8 | 20.8 | 10.6 | 32.8 | 6.4 | 60.2 | 28.0 | 78.6 | **92.6**$_{\pm 2.5}$ |
| antmaze-giant-navigate | 0.2 | 4.0 | **26.2** | 0.0 | 0.0 | 0.0 | 0.0 | 3.8 | 2.6 | 8.6 | 23.0$_{\pm 5.1}$ |
| humanoidmaze-medium-navigate | 2.0 | 32.8 | 21.8 | 0.8 | 1.4 | 52.8 | 19.4 | 38.4 | 60.4 | 57.4 | **75.6**$_{\pm 3.6}$ |
| humanoidmaze-large-navigate | 0.4 | 2.4 | 2.6 | 0.6 | 0.2 | 0.6 | 0.2 | 2.2 | **11.0** | 4.2 | 8.3$_{\pm 5.5}$ |
| antsoccer-arena-navigate | 1.0 | 8.4 | 0.0 | 11.8 | 1.0 | 1.8 | 12.4 | 16.0 | 33.2 | 60.2 | **67.7**$_{\pm 2.9}$ |
| cube-single-play | 5.4 | 83.0 | 90.6 | 94.6 | 79.6 | 85.2 | 81.2 | 78.6 | 79.2 | 95.8 | **98.8**$_{\pm 1.4}$ |
| cube-double-play | 1.6 | 6.4 | 12.2 | 14.6 | 1.4 | 5.8 | 5.2 | 15.0 | 14.0 | 28.6 | **33.1**$_{\pm 5.7}$ |
| scene-play | 4.6 | 27.6 | 40.6 | 46.2 | 20.0 | 39.8 | 29.8 | 44.8 | 30.4 | 55.8 | **71.3**$_{\pm 7.6}$ |
| puzzle-3x3-play | 1.8 | 9.0 | 21.6 | 10.4 | 17.8 | 19.4 | 6.4 | 14.0 | 19.0 | 29.6 | **100.0**$_{\pm 0.0}$ |
| puzzle-4x4-play | 0.2 | 7.4 | 14.0 | 29.2 | 10.6 | 14.8 | 0.4 | 13.2 | 25.2 | 17.2 | **32.3**$_{\pm 6.5}$ |
| visual manipulation (pixel-based) | - | 41.6 | 59.8 | - | - | - | - | 22.8 | 50.2 | 65.4 | **76.0**$_{\pm 3.9}$ |
| **Average (state-based)** | 2.8 | 23.4 | 31.0 | 22.9 | 14.3 | 25.3 | 16.1 | 28.6 | 30.3 | 43.6 | **60.3** |

# 6 EXPERIMENTS

## 6.1 EXPERIMENTAL SETUP

**Benchmarks.** We evaluate FAC on two offline RL benchmarks: recently introduced OGBench (Park et al., 2024c) and standard D4RL (Fu et al., 2020). OGBench offers diverse goal-conditioned locomotion and manipulation environments, in which each task category consists of 5 singletask variants, which are relatively more challenging than the D4RL tasks. To align with standard offline RL, we use 55 reward based singletask variants and follow the 11 OGBench task categories. For D4RL, we use 9 MuJoCo tasks, 6 Antmaze tasks, and 8 Adroit tasks. The details are described in Appendix G.1.

**Baselines.** We evaluate FAC against a total of 17 baselines and categorize them by policy class: Gaussian, diffusion, and flow policies. For OGBench, we compare against Gaussian policy-based baselines BC, IQL (Kostrikov et al., 2021), ReBRAC (Tarasov et al., 2023), diffusion policy-based baselines IDQL (Hansen-Estruch et al., 2023), SRPO (Chen et al., 2024), CAC (Ding & Jin, 2024), and flow policy-based baselines FAWAC, FBRAC, IFQL, FQL (Park et al., 2025). For D4RL, we additionally compare against Gaussian policy-based baselines TD3+BC (Fujimoto & Gu, 2021), CQL (Kumar et al., 2020), MCQ (Lyu et al., 2022), EPQ (Yeom et al., 2024), SAC-RND (Nikulin et al., 2023), SPOT (Wu et al., 2022), and a flow policy-based baseline QIPO-OT (Zhang et al., 2025). The details of the baselines are described in Appendix G.2.

**Evaluation and hyperparameters.** We report the final performance evaluated after 1M gradient steps for state-based tasks and 500K gradient steps for pixel-based tasks. For evaluation, the one-step flow policy trained by FAC samples one action at each time step. For training FAC, we mainly tune two hyperparameters: the critic penalization coefficient $\alpha$ and the actor regularization coefficient $\lambda$. All remaining settings (e.g., dataset-driven threshold schemes) are fixed per task domain. The complete configuration is in Appendix G.3.

## 6.2 PERFORMANCE ON THE OGBENCH BENCHMARK

The evaluation results aggregated from 50 state-based and 5 pixel-based singletasks of the OGBench are summarized in Table 1, and the full evaluation results are in Table 3 in Appendix F. FAC achieves the highest overall performance across all methods. FAC yields a clear margin, compared to FQL among flow policy-based baselines, CAC among diffusion policy-based baselines, ReBRAC among Gaussian policy-based baselines. In particular, the comparison with Gaussian policy-based baselines highlights the benefit of expressive policies on datasets with multi-modal action distribution, while the gap to diffusion policy-based baselines and multi-step flow policy-based baselines (FAWAC, FBRAC, IFQL) reveals the practical difficulty of optimizing iterative generators. Leveraging a one-step flow actor enables both FAC and FQL to outperform other baselines. Furthermore, fusing flow-based critic penalization and actor regularization, FAC significantly improves performance over FQL, which is the actor regularization only.

Table 2: Evaluation on 23 tasks of the D4RL. We report the final performance averaged over 8 seeds, with $\pm$ indicating the standard deviation. For MuJoCo datasets, we use the following abbreviations: *m* for *medium*, *mr* for *medium-replay*, *me* for *medium-expert*.

| | Gaussian Policies | | | | | | | Diffusion Policies | | | Flow Policies | | |
|---|---|---|---|---|---|---|---|---|---|---|---|---|---|
| **MuJoCo Tasks** | TD3+BC | IQL | CQL | MCQ | EPQ | SPOT | ReBRAC | IDQL | SRPO | CAC | QIPO-OT | FQL | FAC (Ours) |
| halfcheetah-m | 48.3 | 47.4 | 44.0 | 64.3 | 67.3 | 58.4 | 65.6 | 51.0 | 60.4 | **69.1** | 54.2 | $60.3_{\pm1.1}$ | $65.0_{\pm1.5}$ |
| hopper-m | 59.3 | 66.3 | 58.5 | 78.4 | 101.3 | 86.0 | **102.0** | 65.4 | 95.5 | 80.7 | 94.1 | $68.1_{\pm3.4}$ | $91.9_{\pm3.9}$ |
| walker2d-m | 83.7 | 78.3 | 72.5 | **91.0** | 87.8 | 86.4 | 82.5 | 82.5 | 84.4 | 83.1 | 87.6 | $77.2_{\pm2.5}$ | $85.2_{\pm0.9}$ |
| halfcheetah-mr | 44.6 | 44.2 | 45.5 | 56.8 | **62.0** | 52.2 | 51.0 | 45.9 | 51.4 | 58.7 | 48.0 | $49.3_{\pm0.5}$ | $55.4_{\pm2.7}$ |
| hopper-mr | 60.9 | 94.7 | 95.0 | **101.6** | 97.8 | 100.2 | 98.1 | 92.1 | 101.2 | 99.7 | 101.3 | $49.8_{\pm7.2}$ | $99.1_{\pm0.9}$ |
| walker2d-mr | 81.8 | 73.9 | 77.2 | 91.3 | 85.3 | **91.6** | 77.3 | 85.1 | 84.6 | 79.5 | 78.6 | $53.1_{\pm7.9}$ | $83.0_{\pm5.8}$ |
| halfcheetah-me | 90.7 | 86.7 | 87.5 | 87.5 | 95.7 | 86.9 | 101.1 | 95.9 | 92.2 | 84.3 | 94.5 | $99.6_{\pm6.5}$ | $\mathbf{101.9}_{\pm5.6}$ |
| hopper-me | 98.0 | 91.5 | 105.4 | **111.2** | 108.8 | 99.3 | 107.0 | 108.6 | 100.1 | 100.4 | 108.0 | $83.1_{\pm17.0}$ | $104.2_{\pm4.6}$ |
| walker2d-me | 110.1 | 109.6 | 108.8 | **114.2** | 112.0 | 112.0 | 111.6 | 112.7 | 114.0 | 110.4 | 110.9 | $106.1_{\pm1.8}$ | $108.4_{\pm0.6}$ |
| **Average** | 75.3 | 77.0 | 77.6 | 88.5 | **90.9** | 85.9 | 88.5 | 82.1 | 87.1 | 85.1 | 86.4 | 71.8 | 88.2 |

| | Gaussian Policies | | | | | | | Diffusion Policies | | | Flow Policies | | |
|---|---|---|---|---|---|---|---|---|---|---|---|---|---|
| **Antmaze Tasks** | TD3+BC | IQL | CQL | MCQ | EPQ | SAC-RND | ReBRAC | IDQL | SRPO | CAC | QIPO-OT | FQL | FAC (Ours) |
| umaze | 78.6 | 87.5 | 74.0 | 98.3 | **99.4** | 97.0 | 97.8 | 94.0 | 97.1 | 75.8 | 93.6 | 96.0 | $98.5_{\pm3.0}$ |
| umaze-diverse | 71.4 | 62.2 | 84.0 | 80.0 | 78.3 | 66.0 | 88.3 | 80.2 | 82.1 | 77.6 | 76.1 | 89.0 | $\mathbf{93.5}_{\pm6.0}$ |
| medium-play | 10.6 | 71.2 | 61.2 | 52.5 | 85.0 | 38.5 | 84.0 | 84.5 | 80.7 | 56.8 | 80.0 | 78.0 | $\mathbf{88.0}_{\pm9.6}$ |
| medium-diverse | 3.0 | 70.0 | 53.7 | 37.5 | **86.7** | 74.7 | 76.3 | 84.8 | 75.0 | 0.0 | 86.4 | 71.0 | $85.0_{\pm7.3}$ |
| large-play | 0.2 | 39.6 | 15.8 | 2.5 | 40.0 | 43.9 | 60.4 | 63.5 | 53.6 | 0.0 | 55.5 | 84.0 | $\mathbf{90.0}_{\pm4.3}$ |
| large-diverse | 0.0 | 47.5 | 14.9 | 7.5 | 36.7 | 45.7 | 54.4 | 67.9 | 53.6 | 0.0 | 32.1 | 83.0 | $\mathbf{88.0}_{\pm6.0}$ |
| **Average** | 27.3 | 63.0 | 50.6 | 46.4 | 71.0 | 61.0 | 76.9 | 79.2 | 73.7 | 35.0 | 70.6 | 83.5 | **90.5** |

| | Gaussian Policies | | | | | | | Diffusion Policies | | | Flow Policies | |
|---|---|---|---|---|---|---|---|---|---|---|---|---|
| **Adroit Tasks** | TD3+BC | IQL | CQL | MCQ | EPQ | SAC-RND | ReBRAC | IDQL | SRPO | CAC | FQL | FAC (Ours) |
| pen-human | 81.8 | 81.5 | 37.5 | 68.5 | 83.9 | 5.6 | **103.5** | 76.0 | 69.0 | 64.0 | 53.0 | $73.9_{\pm14.7}$ |
| pen-cloned | 61.4 | 77.2 | 39.2 | 49.4 | 91.8 | 2.5 | 91.8 | 64.0 | 61.0 | 56.0 | 74.0 | $\mathbf{103.2}_{\pm11.1}$ |
| door-human | -0.1 | 3.1 | 9.9 | 2.3 | **13.2** | 0.0 | 0.0 | 6.0 | 3.0 | 5.0 | 0.0 | $5.5_{\pm3.3}$ |
| door-cloned | 0.1 | 0.8 | 0.4 | 1.3 | **5.8** | 0.2 | 1.1 | 0.0 | 0.0 | 1.0 | 2.0 | $4.1_{\pm3.9}$ |
| hammer-human | 0.4 | 2.5 | 4.4 | 0.3 | 3.9 | -0.1 | 0.2 | 2.0 | 1.0 | 2.0 | 1.0 | $\mathbf{8.6}_{\pm5.4}$ |
| hammer-cloned | 0.8 | 1.1 | 2.1 | 1.4 | **22.8** | 0.1 | 6.7 | 2.0 | 2.0 | 1.0 | 11.0 | $11.1_{\pm11.2}$ |
| relocate-human | -0.2 | 0.1 | 0.2 | 0.1 | 0.3 | 0.0 | 0.0 | 0.0 | 0.0 | 0.0 | 0.0 | $\mathbf{0.6}_{\pm0.5}$ |
| relocate-cloned | -0.1 | 0.2 | -0.1 | 0.0 | 0.1 | 0.0 | **0.9** | 0.0 | 0.0 | 0.0 | 0.0 | $0.5_{\pm0.4}$ |
| **Average** | 18.0 | 20.8 | 11.7 | 15.4 | **27.7** | 1.0 | 25.5 | 18.8 | 17.0 | 16.1 | 17.6 | 25.9 |

On navigation tasks, FAC achieves the best performance on 3 out of 5 navigation tasks. Although `antmaze-giant-navigate` remains led by ReBRAC, indicating that a joint critic penalization and actor regularization approach can be competitive even with Gaussian policies. For higher-dimensional `humanoid` tasks, FAC outperforms ReBRAC, suggesting that the expressive actor contributes in complex action spaces. In `humanoidmaze-large-navigate`, FAC is slightly below IFQL on average but matches or exceeds this baseline on 4 of the 5 singletasks (see Table 3 in the appendix for details.).

On state-based and pixel-based manipulation tasks, FAC outperforms all baselines across the board and achieves perfect success on `puzzle-3x3-play`. In particular, the `puzzle-play` tasks for solving lights-out puzzle are designed to test combinatorial reasoning and trajectory stitching, on these tasks, FAC attains 238% and 11% performance improvements over the strongest baselines (FQL and IDQL) on `puzzle-3x3-play` and `puzzle-4x4-play`, respectively. Moreover, FAC achieves the highest performance on the pixel-based tasks, demonstrating that performance gains of the proposed method also hold in these higher-dimensional settings.

## 6.3 PERFORMANCE ON THE D4RL BENCHMARK

The evaluation results on 23 tasks of the D4RL are summarized in Table 2. Overall, FAC is competitive or superior across all three domains (MuJoCo, Antmaze, Adroit), indicating that the benefit of jointly penalizing the critic and regularizing the actor with the expressive behavior proxy policy extends beyond the OGBench and holds under diverse reward functions and state-action distributions. Among methods with expressive policies, FAC attains the best overall performance across all three domains.

On MuJoCo tasks with dense reward functions, it is susceptible to value overestimation for OOD actions. It is seen that FQL exhibits sensitivity to the value overestimation bias, while FAC matches

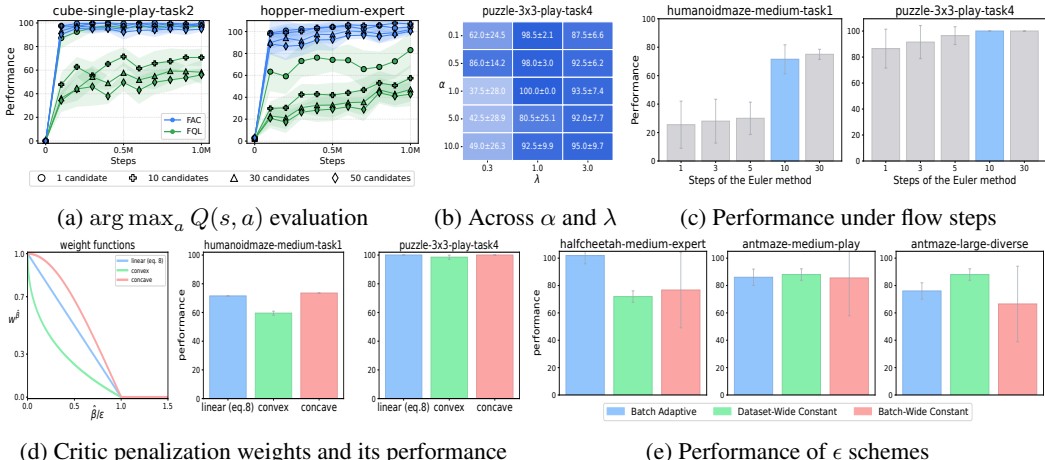

Figure 3: Empirical analysis of components. (a) Effect of flow-based critic penalization (CP) on performance under $N$ candidate actions. (b) Performances across actor regularization coefficient $\lambda$ and critic penalization coefficient $\alpha$. (c) Influence by the fidelity of flow behavior proxy under flow steps $T$. (d) Performance of weight function designs. (e) Performance under different $\epsilon$ schemes.

strong Gaussian policy-based baselines. This demonstrates that the flow-based critic penalization in FAC successfully mitigates the value overestimation.

On Antmaze tasks, as task difficulty increases from `umaze` to `medium` to `large`, many baselines including the strong Gaussian policy-based baselines on the MuJoCo domain, diffusion policy-based baselines, and the multistep flow policy-based baseline exhibit pronounced performance degradation. In contrast, FAC maintains robust performance against difficulty and consistently improves over FQL.

On Adroit tasks that are notoriously challenging due to the high-dimensional action space and the sparse dataset support, FAC is comparable to the strong Gaussian policy-based baselines in overall performance. In particular, FAC attains substantial improvements on `pen-cloned` and `hammer-human` tasks.

## 6.4 EMPIRICAL ANALYSIS

**Effect of the critic penalization.** We investigate the impact of the proposed flow-based critic penalization of FAC in terms of OOD action sampling by replacing our flow-based penalized critic with an unpenalized conventional Q estimator (i.e., reducing to FQL). For this, we sample $N \in \{1, 10, 30, 50\}$ candidate actions $a(s, z)$ from the one-step flow actor with $N$ i.i.d. $z$ for given $s$, and execute $\arg\max_a Q_\phi(s, a)$ at each step for each method, with its best-performing configurations for $\lambda$, $\alpha$, and flow behavior proxy model. Note that as $N$ increases, the chance of selecting an OOD action increases and $\arg\max_a Q_\phi(s, a)$ with the unpenalized $Q$ can check this. As shown in Fig. 3a, as $N$ increases, FQL exhibits significant performance degradation, showing that OOD actions are drawn even with the distance regularization. In contrast, our method maintains consistent performance with increasing $N$, suggesting that the flow-based critic penalization together with actor regularization is effective for handling potential OOD actions.

**Sensitivity to $\alpha$ and $\lambda$.** FAC substantially improves performance on `puzzle-3x3-play-v0`. Fig. 3b shows a performance heatmap across critic penalization coefficient $\alpha$ and actor regularization coefficient $\lambda$ on this default singletask, with each cell presenting the mean $\pm$ standard deviation over 8 seeds. With adequate $\lambda \geq 1$, performance is robust over $\alpha$, whereas with too small $\lambda = 0.3$, performance becomes sensitive to $\alpha$. These results demonstrate that jointly applying flow-based critic penalization and flow actor regularization produces a synergistic effect, enabling reliable support-aware flow policy optimization in the offline RL setting.

**Influence by the fidelity of flow behavior proxy.** The step count of the Euler method directly affects the fidelity of the approximated flow obtained by the flow proxy. When the learned flow represents almost linear trajectories from base noises $x_0$ to data samples $x_1$, single step count would be sufficient.

However, for complex data distributions where the learned flow exhibit curved trajectories, higher curvature needs more integration steps (refer to Fig. 5 in the appendix). As shown in Fig. 3c, FAC attains high performance for $T \geq 10$, demonstrating that sufficient fidelity of the flow behavior proxy is essential for the strong performance of FAC.

**Impact of critic penalization weight functions.** We further investigate alternative designs for the critic penalization weight $w^{\hat{\beta}_\psi}(s, a)$. For clarity, we refer to the original weight (eq. 8) as linear weight and introduce two alternatives defined as $\max(0, 1 - (\log(\hat{\beta}_\psi(a|s)/\epsilon)^\kappa + 1)/\log 2)$, here $\kappa = 2$ and $\kappa = 1/2$ yield concave and convex weight functions, respectively, on $[0, 1]$ of $\hat{\beta}_\psi/\epsilon$. Fig. 3d presents the weight functions and the performance of FAC under the weights. For the tasks, the linear and concave weights achieve comparable performance, however, the convex weight leads to performance degradation. These results indicate that the linear weight is a simple but effective choice.

**Impact of $\epsilon$ schemes.** The choice of $\epsilon$ is an important component of the critic penalization weight (eq. 8). We evaluate three dataset-driven threshold designs: (i) batch adaptive, (ii) dataset wide constant, and (iii) newly introduced batch wide constant, $\min_{(s,a)\sim\mathcal{B}} \hat{\beta}_\psi(a|s)$, where $\mathcal{B}$ denotes a mini-batch. Fig. 3e shows that the three $\epsilon$ schemes yield distinct performance on `halfcheetah-medium-expert-v2` and `antmaze-large-diverse-v2`, whereas they exhibits nearly identical performance on `antmaze-medium-play-v2`. Based on these observations, we set $\epsilon$ as batch adaptive for all MuJoCo tasks and as dataset wide constant for all Antmaze tasks, in order to avoid excessive $\epsilon$ selection.

**Online fine-tuning.** We evaluate FAC in the offline-to-online setting. Fig. 4 presents the results in this setting, where the first 1M gradient steps denote the offline RL training phase, and the subsequent 1M steps denote the online finetuning phase. We note that FAC reuses the hyperparameters used in the offline RL settings. The details for online finetuning are provided in Appendix G.3.3.

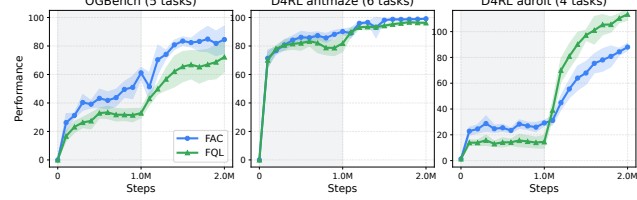

Figure 4: Offline-to-online results, averaged over 8 seeds with standard deviation. The gray shaded region indicates the offline phase, and the online fine-tuning phase is left unshaded. Full plots on 15 tasks are in Fig. 7.

We compare FAC with FQL, which is also a competitive baseline. On the OGBench and D4RL Antmaze tasks, FAC achieves strong performance even in the offline-to-online setting. These results may come from the proposed flow-based critic penalization design. The critic penalization design allows to reduce or eliminate penalization for confident actions, suggesting that FAC avoids excessively suppressing high Q-value neighborhoods near the dataset support boundary. This, in turn, may enable exploration in such high Q-value regions. However, on the D4RL Adroit tasks, FAC shows smaller performance gains during online finetuning, despite outperforming FQL in offline training. Due to the notoriously sparsity of the Adroit datasets, FAC during the offline training phase provides limited guidance about high Q-value regions, and the critic penalization may unintentionally hinder exploration toward these promising neighborhoods.

# 7 CONCLUSION

We have proposed *flow actor-critic* (FAC), which leverages a flow behavior proxy policy jointly for critic penalization and actor regularization. The behavior proxy provides tractable behavior densities that yield confidence weights for critic penalization so that the resulting operator preserves the original Bellman operator in confident in-distribution region and penalizes value overestimation for out-of-distribution actions in complex and multimodal datasets. This behavior proxy also regularizes the flow actor, constraining it to the dataset support. Consequently, this joint approach enables flow-based actor-critic optimization that is reliably constrained within well-supported regions. Empirically, FAC achieves consistently strong performance across tasks of the OGBench and D4RL benchmarks, effectively handling out-of-distribution actions and maintaining high true returns under diverse reward structures and state-action distributions.

## REPRODUCIBILITY STATEMENT

Our code is available at `https://github.com/JongseongChae/FAC`. The details of the practical implementation and experimental setup are described in Section 4.3 and Appendix G.3.

## ACKNOWLEDGMENTS

This work was supported in part by the Institute of Information & Communications Technology Planning & Evaluation (IITP) grant funded by the Korea government (MSIT) (No.RS-2022-II220469, Development of Core Technologies for Task-oriented Reinforcement Learning for Commercialization of Autonomous Drones) and in part by Institute of Information & Communications Technology Planning & Evaluation (IITP) grant funded by the Korea government (MSIT) (No. RS-2024-00457882, AI Research Hub Project)

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

# A    LIMITATIONS

The proposed method relies on the fidelity and accuracy of density evaluation of the flow behavior proxy policy for the underlying behavior policies. When the behavior proxy fails to capture the multi-modal dataset distribution, the critic penalization with the flow behavior density can suppress values in inappropriate regions, and the distance regularization for flow actor optimization can drive the flow actor toward the regions. In addition to this sensitivity, the method provides no mechanism for beneficial exploration, leaving its applicability to online RL uncertain. These limitations suggest future extensions such as robust learning of the behavior proxy (Liu et al., 2022) to enable reliable support-aware policy improvement, and efficient exploitation strategies (Haarnoja et al., 2018; Shin et al., 2026) for flow policies.

Moreover, our evaluation is limited to simulated benchmarks, and we have not evaluated the method on real-world tasks. Real-world deployment introduces additional limitations, such as inherent stochasticity in environment dynamics (Morimoto & Doya, 2005), the possibility of leveraging data from different domains due to data scarcity or biased data collection (Seo et al., 2024), and the need to balance multiple conflicting objectives in reward design (Yang et al., 2019). Thus, extending the method to robust RL (Pinto et al., 2017; Wang & Zou, 2022; Chae et al., 2022), domain adaptation (Gupta et al., 2017; Choi et al., 2023), and multi-objective RL (Park et al., 2024a; Byeon et al., 2025) to handle these real-world challenges remains an important direction for future work.

# B    THE USE OF LARGE LANGUAGE MODELS

We used large language models to refine and polish our writing.

## C  BEHAVIOR CLONING MODELS

In Fig. 1, we compare the flow behavior proxy model against other behavior cloning (BC) models. Specifically, we investigate data sampling as well as log-density or Evidence Lower Bound (ELBO) evaluations when employing a flow matching model (Lipman et al., 2022), a simple Gaussian probability model, a conditional VAE (Kingma & Welling, 2013; Sohn et al., 2015), and diffusion models with $T = \{10, 50\}$ denoising steps (Ho et al., 2020).

### C.1  FLOW BEHAVIOR PROXY MODEL UNDER DIFFERENT STEPS OF NUMERICAL INTEGRATION SOLVER

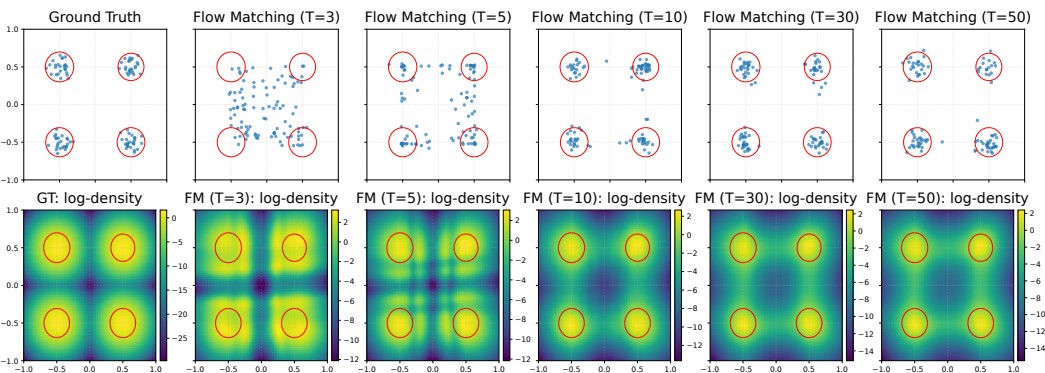

Figure 5: Flow behavior proxy model with different step counts of the Euler method.

We investigate how the fidelity of the flow behavior proxy affects both sampling quality and density estimation. The flow behavior proxy is trained by flow matching objective 6, meaning that we can use different step counts of numerical integration for both data sampling and density estimation. We use the Euler method as the numerical integration solver, and the step counts directly influence the fidelity of the learned flow. When the flow matching model learns a nearly linear trajectory from base noise $x_0$ to data sample $x_1$, the single step count would be sufficient. However, when the underlying data distribution is complex and the learned trajectory exhibits curve, higher curvature requires larger step counts of the Euler method for sufficient sampling and density estimation.

We evaluate sampling and density estimation quality under different Euler step count $T \in \{3, 5, 10, 30, 50\}$. Figure 5 shows the generated samples (top row) and log-density estimates (bottom row) across different $T$. When $T$ is below 10, sampling quality degrades significantly, and density estimation with smallest $T = 3$ fails to separate ID and OOD regions, making it unsuitable for use in the proposed critic penalization. In contrast, for $T \geq 10$, both sampling and density estimation reach a level of fidelity with sufficient recovery of the multimodal distribution and accurate density contrast between ID and OOD regions. The results demonstrate that a moderate number of Euler step (e.g., $T = 10$) achieves strong fidelity required for density-based critic penalization.

### C.2  IMPLEMENTATION

**Dataset.** We construct a dataset by generating a single state, $s = [0.5, -0.5, 0.5, -0.5]$, and 2-dimensional actions from a Gaussian mixture with four modes, $a \sim \frac{1}{4} \sum_{i=1}^{4} \mathcal{N}(\mu_i, \Sigma = 0.008I)$, where $\mu_i \in \{[-0.5, -0.5], [-0.5, 0.5], [0.5, -0.5], [0.5, 0.5]\}$.

**Network architecture.** We use a $[512, 512, 512, 512]$-sized MLP for all neural networks. For the flow matching model, we implement one neural network: velocity prediction network that takes the state $s$, Gaussian noise $z \in \mathbb{R}^{d_\mathcal{A}}$, and flow time $u \in [0, 1]$ and then outputs velocity estimate $v_u \in \mathbb{R}^{d_\mathcal{A}}$. For the Gaussian model, we use one neural network: Gaussian probability model, $\mathcal{N}(\cdot|\mu_\theta, \sigma_\theta)$. For the conditional VAE, we implement two neural networks: (1) Encoder network that takes the state $s$ and one action $a$ then outputs latent variable $z$, and (2) Decoder network that takes the state $s$ and latent variable $z$ then outputs an action estimate. For the diffusion models, we use one neural network:

conditional $\epsilon$ prediction network that takes the state $s$, diffusion time $t$, and latent variable $x \in \mathbb{R}^{d_\mathcal{A}}$ and then outputs $\epsilon$-estimate.

## C.3 TRAINING OBJECTIVE AND ACTION SAMPLING

**Flow Matching model.** We train a flow matching model with the flow matching objective (6) and restate it:

$$\min_\psi \ \mathbb{E}_{(s,a)\sim\mathcal{D},z\sim\mathcal{N}(0,I),u\sim\text{Unif}([0,1])} \left[ \|v_\psi(\tilde{a}_u; s, u) - (a - z)\|_2^2 \right]$$

For action sampling, we solve the following integral equation using the Euler method with 10 steps:

$$a = z + \int_0^1 v_u du = z + \int_0^1 v_\psi(a_u; s, u) du,$$

where $a_u = \int_0^u v_t dt$, and $z \sim \mathcal{N}(0, I)$.

**Gaussian model.** A Gaussian model is trained by maximizing log-likelihood:

$$\max_\theta \ \mathbb{E}_{s,a\sim\mathcal{D}} \left[ \log p(a|s) \right] = \mathbb{E}_{s,a\sim\mathcal{D}} \left[ \log \mathcal{N}(a|\mu_\theta(s), \sigma_\theta(s)) \right]$$

We can easily sample actions from the Gaussian model.

**Conditional VAE model.** We train Encoder and Decoder models, $q_\phi(z|s,a)$ and $p_\theta(a|s,z)$, respectively, by minimizing the ELBO of log-likelihood (Kingma & Welling, 2013). The objective is given by

$$\max_{\theta,\phi} \ \mathbb{E}_{s,a\sim\mathcal{D}} \left[ \log p(a|s) \right] \geq \mathbb{E}_{s,a\sim\mathcal{D}} \left[ \mathbb{E}_{z\sim q_\phi} \left[ \log p_\theta(a|s,z) \right] - D_{\text{KL}}(q_\phi(z|s,a)\|p(z)) \right],$$

where the prior distribution $p(z)$ is set to $\mathcal{N}(0, I)$.

For action sampling, we sample $z \sim \mathcal{N}(0, I)$, and feed the state $s$ and $z$ to the Decoder model.

**Diffusion model.** We train an $\epsilon$-prediction model implemented as in DDPM (Ho et al., 2020).

$$\min_{\epsilon_\theta} \mathbb{E}_{t\sim\text{Unif}(\{1,...,T\}),\epsilon\sim\mathcal{N}(0,I),(s,a)\sim\mathcal{D}} \left[ \|\epsilon - \epsilon_\theta(\sqrt{\bar{\alpha}_t}a + \sqrt{1 - \bar{\alpha}_t}\epsilon, s, t)\|^2 \right], \tag{12}$$

where $\alpha_t = 1 - \beta_t$, and $\bar{\alpha}_t = \prod_{s=1}^t \alpha_s$. Following (Xiao et al., 2021) as in (Wang et al., 2022), we use the beta schedule $\beta_t = 1 - \exp\left(-\frac{\beta_{\min}}{N} - \frac{(\beta_{\max}-\beta_{\min})(2t-1)}{2N^2}\right)$.

For action sampling, we first sample a noise $a_T \sim \mathcal{N}(0, I)$, and then iteratively generate actions through the reverse process of the diffusion model:

$$a_{t-1}|a_t = \frac{a_t}{\sqrt{\alpha_t}} - \frac{\beta_t}{\sqrt{\alpha_t(1 - \bar{\alpha}_t)}}\epsilon_\theta(s, a_t, t) + \sqrt{\beta_t}\epsilon,$$

where $\epsilon \sim \mathcal{N}(0, I)$. Note that in the diffusion model, the denoising time $t = 0$ corresponds to the data index $a_0$, while $t = T$ corresponds to the noise index $a_T$, which is the opposite of the indexing convention in flow-based models.

## C.4 LOG-DENSITY AND ELBO FOR ARBITRARY ACTIONS

**Flow Matching model.** We can directly evaluate the log-density using the Instantaneous Change of Variables (Chen et al., 2018), which is a property of Continuous Normalizing Flows.

$$\log p(a|s) = \log p_0(\hat{z}) + \int_0^1 \frac{d}{du} \log p_u(a|s) du = \log p_0(\hat{z}) - \int_0^1 \nabla_{a_u} \cdot v_\psi(a_u; s, u) du,$$

where $a_u = a - \int_u^1 v_t dt$, and the base distribution $p_0$ is set to normal distribution. We also use the 10 step count of the Euler method. Note that $\hat{z}$ is obtained from $a = a_1$ due to the bijective property of Continuous Normalizing Flows.

**Gaussian model.** We can easily compute log-density.

$$\log p(a|s) = \log \mathcal{N}(a|\mu_\theta(s), \sigma_\theta(s))$$

**Conditional VAE model.** Since the VAE model cannot compute log-density exactly, we instead evaluate ELBO, which is a lower bound of its log-density. Using a normal distribution as the prior $p(z)$ and implementing both the encoder $q_\phi$ and decoder $p_\theta$ as Gaussian probability models, ELBO can be computed in a straightforward manner:

$$\log p(a|s) \geq \mathbb{E}_{z \sim q_\phi}\left[\log p_\theta(a|s,z)\right] - D_{\mathrm{KL}}(q_\phi(z|s,a)\|p(z))$$

**Diffusion model.** Diffusion model (Sohl-Dickstein et al., 2015) defines a forward process that gradually perturbs data to noise through a Markov chain, and a reverse process that learns to reconstruct data by denoising procedure that inverts the forward process. In the diffusion model, its log-likelihood is optimized via a variational lower bound (ELBO).

DDPM (Ho et al., 2020) simplifies the ELBO by reparameterizing the reverse process as the $\epsilon$-prediction objective (12), which allows the ELBO to be expressed in a tractable form while retaining its variational interpretation.

In the RL setting, we define the forward process $q(a_t|a_{t-1},s) = \mathcal{N}(a_t; \sqrt{1-\beta_t}a_{t-1}, \beta_t I)$ and the reverse process $p_\theta(a_{t-1}|a_t,s) = \mathcal{N}(a_{t-1}; \mu_\theta(s,a_t,t), \Sigma_\theta(s,a_t,t))$, following (Sohl-Dickstein et al., 2015). We therefore compute the ELBO as follows:

$$\begin{aligned}
\log p(a|s) &\geq \mathbb{E}_q\left[\log \frac{p_\theta(a_{0:T}|s)}{q(a_{1:T}|a_0,s)}\right] \\
&= \mathbb{E}_q\Big[\log p_\theta(a_0|a_1,s) - \sum_{t>1} D_{\mathrm{KL}}(q(a_{t-1}|a_t,a_0,s)\|p_\theta(a_{t-1}|a_t,s)) \\
&\quad - D_{\mathrm{KL}}(q(a_T|a_0,s)\|p(a_T))\Big],
\end{aligned} \tag{13}$$

where $p(a_T)$ is a normal distribution.

We already have the predefined beta schedule $\{\beta_t\}$ and the trained $\epsilon$-prediction model, we can choose the forward process as in DDPM (Ho et al., 2020):

$$p_\theta(a_{t-1}|a_t,s) = \mathcal{N}(a_{t-1}; \mu_\theta(s,a_t,t), \tilde{\beta}_t I),$$

where $\tilde{\beta}_t = \frac{1-\bar{\alpha}_{t-1}}{1-\bar{\alpha}_t}\beta_t$.

Using notable properties of the diffusion model (Sohl-Dickstein et al., 2015)

$$q(a_t|a_0,s) = \mathcal{N}\left(a_t; \sqrt{\bar{\alpha}_t}a_0, (1-\bar{\alpha}_t)I\right)$$

$$q(a_{t-1}|a_t,a_0,s) = \mathcal{N}\left(a_{t-1}; \tilde{\mu}_t(a_t,a_0), \tilde{\beta}_t I\right),$$

where $\tilde{\mu}_t(a_t,a_0) = \frac{\sqrt{\bar{\alpha}_{t-1}}\beta_t}{1-\bar{\alpha}_t}a_0 + \frac{\sqrt{\alpha_t}(1-\bar{\alpha}_{t-1})}{1-\bar{\alpha}_t}a_t$, and the KL-divergence between two multivariate Gaussian distributions, $D_{\mathrm{KL}}(\mathcal{N}(\mu_1,\Sigma_1)\|\mathcal{N}(\mu_2,\Sigma_2))$,

$$D_{\mathrm{KL}}(\mathcal{N}_1\|\mathcal{N}_2) = \frac{1}{2}\left\{\mathrm{tr}\left(\Sigma_2^{-1}\Sigma_1\right) + (\mu_2-\mu_1)^\top \Sigma_2^{-1}(\mu_2-\mu_1) - d_{\mathcal{A}} + \log\frac{\det(\Sigma_2)}{\det(\Sigma_1)}\right\},$$

where $d_{\mathcal{A}}$ denotes the dimension of the action space, each term of the ELBO (13) can be computed as

$$\log p_\theta(a_0|a_1,s) = -\frac{d_{\mathcal{A}}}{2}\log(2\pi\tilde{\beta}_1) - \frac{1}{2\tilde{\beta}_1}\|a_0 - \mu_\theta(s,a_1,1)\|^2$$

$$D_{\mathrm{KL}}(q(a_{t-1}|a_t,a_0,s)\|p_\theta(a_{t-1}|a_t,s)) = \frac{1}{2\tilde{\beta}_t}\|\tilde{\mu}_t(a_t,a_0) - \mu_\theta(s,a_t,t)\|^2,$$

$$D_{\mathrm{KL}}(q(a_T|a_0,s)\|p(a_T)) = \frac{1}{2}\left(\bar{\alpha}_T a_0^\top a_0 - d_{\mathcal{A}}(\bar{\alpha}_T + \log(1-\bar{\alpha}_T))\right)$$

where $\mu_\theta(s,a_t,t) = \frac{1}{\sqrt{\alpha_t}}\left(a_t - \frac{\beta_t}{\sqrt{1-\bar{\alpha}_t}}\epsilon_\theta(s,a_t,t)\right)$.

# D    LEARNED Q COMPARISON ON A CONTINUOUS-ACTION BANDIT PROBLEM

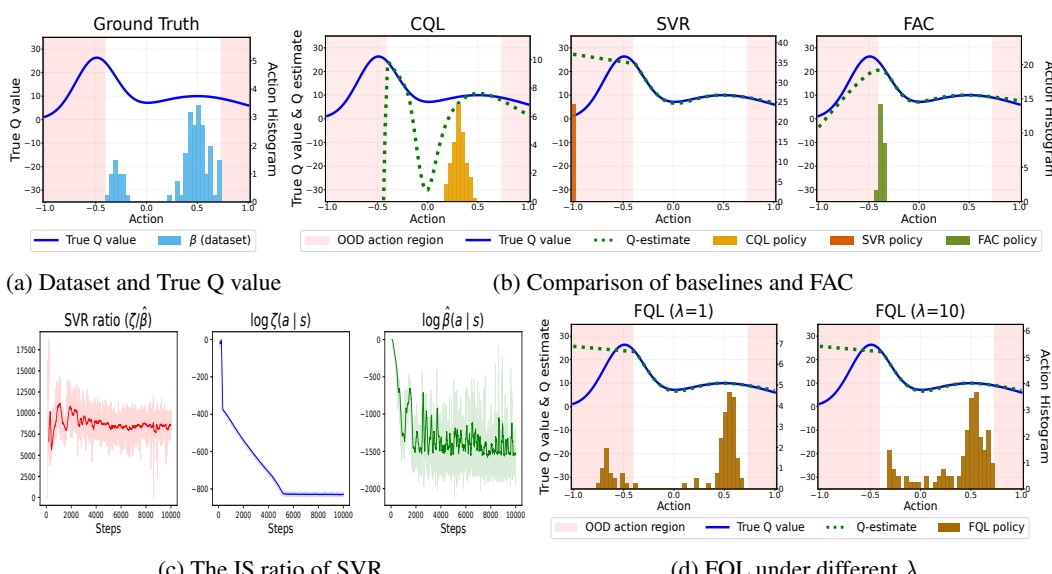

Figure 6: A comparison of FAC and baselines in a continuous action bandit problem. (a) Dataset and true Q value of the problem. (b) Comparison of baselines and FAC. (c) The IS ratio in the critic penalization of SVR. (d) Results of FQL under two actor regularization coefficients $\lambda = \{1, 10\}$.

In order to compare the accuracy of the learned Q function, we consider a concrete example in this section. We mainly consider CQL (an early work), SVR (a good method with Gaussian policy), FQL (a recent flow-based method) and FAC (our method) for learned Q function comparison.

## D.1    THE CONSIDERED CONTINUOUS-ACTION BANDIT PROBLEM

Since we need to know the true action value function, i.e., Q function, for accuracy evaluation and it is difficult to know the true Q function for complicated RL tasks, we consider a continuous-action bandit problem, which is modified from the bandit example in Ch.2 of Sutton & Barto (2018). In the considered bandit problem, the action range is from -1 to 1, and each time we perform an action to hit the desired points located at -0.5 and 0.5, where the location -0.5 is preferred. Thus, the true action value function is designed as a Gaussian mixture and given by

$$Q(a) = \bar{r}(a) = 25e^{-25(a+0.5)^2/2} + 10e^{-2(a-0.5)^2}.$$

The true action value function is shown in Figure 6a. As seen, the true Q value exhibits two modes: the left mode with high peak and the right mode with a moderately lower value. The reward at each time is a noisy version of the true action value, i.e.,

$$r(a) = \bar{r}(a) + \xi,$$

where $\xi$ is zero-mean Gaussian noise with standard deviation $0.1$. Note that due to the noise in reward, we cannot know the true value from a single trial, and exploration to find a better action and exploitation to use the collected information are necessary for this simple bandit problem to obtain a large cumulated reward over time.

**Dataset generation:** We construct an offline dataset composed of data $\{(s_0, a, r)\}$ sampled from two Gaussian distributions: one centered at $0.5$ and the other centered at $-0.3$, and $80\%$ of data is from the sampling distribution centered at $0.5$ and the remaining $20\%$ of data is from the sampling distribution centered at $-0.3$, as show in Figure 6a. Here, $s_0$ denotes the dummy single state of the bandit problem. The reason for this dataset construction is that we want to model the situation in which a large portion of dataset corresponds to suboptimal behavior and a small portion corresponds to the near-optimal behavior.

## D.2 RESULTS AND COMPARISON

The comparison result is shown in Figure 6 (b) and (d), where the blue solid line is the true Q function and the green dotted lines are the learned Q functions of the considered methods.

**FAC versus Gaussian Model-Based Methods:** We first compare the performance with representative conservative critic methods: CQL and SVR, which are based on Gaussian probability models. Fig. 6b shows the learned critic and actor for each method in the continuous action bandit setting.

The learned critic of CQL significantly underestimates the Q value between the two modes of the dataset, resulting in the actor being confined to the suboptimal ID region. In contrast, SVR tends to overestimate the value in the OOD region. The learned critic overestimates the OOD region, which induces the actor to sample OOD actions. This result stems from the instability of the importance sampling (IS) ratio used in the critic penalization term of SVR, as shown in Fig. 6c. As a consequence of this instability, the penalization term in the SVR loss ceases to play a corrective role, and the learned critic becomes unstable during training[1]. In contrast, FAC provides more accurate Q estimates in ID region and gradually reduces estimates in OOD regions, which leads the actor to concentrate on the well-supported high-value region.

**FAC versus FQL:** Now, we compare FQL, which uses only the actor regularization, and our FAC. As shown in Fig. 6d, the one-step flow actor, which is used in both methods, captures the multi-modal action distribution, confirming sufficient expressibility on this problem. The key difference exists in critic learning. FQL tends to overestimate in the OOD region, which leads to OOD action sampling under weak regularization and to over-imitation of sub-optimal actions under strong regularization, as shown in Fig. 6d. In contrast, FAC preserves more accurate values on the in-distribution (ID) region and gradually suppresses values in the OOD region. Hence, the actor concentrates on the well-supported high-value region even with weaker actor regularization $\lambda = 0.1$, as shown in Fig. 6b. These observations demonstrate the synergy of the combination used in FAC: how flow-based critic penalization complements flow actor regularization by aligning critic learning with dataset support while retaining multi-modal expressibility.

## D.3 ALGORITHMS

We train all methods by adapting them to the considered bandit setting, where $Q_\phi$ is regressed from noisy reward $r$, similarly to Ball et al. (2023). For the flow behavior proxies of FAC and FQL, we use a behavior flow proxy $\hat{\beta}_\psi$ trained via the flow matching objective (6). We use the Euler method with 10 steps as an integral (ODE) solver for action sampling and evaluating behavior proxy density from $\hat{\beta}_\psi$. For the behavior proxy of SVR, we use a Gaussian behavior cloning model $\hat{\beta}_\psi$ trained via maximizing log-likelihood on the dataset. The actor and critic objectives of the methods are presented below.

**FAC.** The objectives of FAC are given by

$$\max_{\pi_\theta} \mathbb{E}_{s\sim\mathcal{D}} \left[ \mathbb{E}_{a\sim\pi_\theta} \left[ Q_\phi(s,a) \right] - \lambda \mathbb{E}_{z\sim\mathcal{N}(0,I)} \left[ \|a_\theta(s,z) - a_\psi(s,z)\|_2^2 \right] \right]$$

$$\min_{Q_\phi} \alpha \mathbb{E}_{s\sim\mathcal{D},\, a\sim\pi} \left[ w^{\hat{\beta}_\psi}(s,a) \cdot Q(s,a) \right] + \mathbb{E}_{s,a\sim\mathcal{D}} \left[ (Q(s,a) - r(a))^2 \right]$$

**FQL.** The objectives of FQL are given by

$$\max_{\pi_\theta} \mathbb{E}_{s\sim\mathcal{D}} \left[ \mathbb{E}_{a\sim\pi_\theta} \left[ Q_\phi(s,a) \right] - \lambda \mathbb{E}_{z\sim\mathcal{N}(0,I)} \left[ \|a_\theta(s,z) - a_\psi(s,z)\|_2^2 \right] \right]$$

$$\min_{Q_\phi} \mathbb{E}_{s,a\sim\mathcal{D}} \left[ (Q(s,a) - r(a))^2 \right]$$

---

[1]To stabilize training, we follow the official open-source implementation of SVR and clip the critic penalization term to the range $[-10^4, 10^4]$. Despite this practical adjustment, the IS ratio still exhibits values near the clipping threshold, limiting the effectiveness of the penalization term.

**CQL.** The objectives of CQL are given by,

$$\max_{\pi_\theta} \; \mathbb{E}_{s\sim\mathcal{D},\, a\sim\pi_\theta} \left[ Q_\phi(s,a) - \eta \log \pi_\theta(a|s) \right]$$

$$\min_{Q_\phi} \; \alpha \, \mathbb{E}_{s\sim\mathcal{D}} \left[ \mathbb{E}_{a\sim\pi_\theta} \left[ Q_\phi(s,a) \right] - \mathbb{E}_{a\sim\mathcal{D}} \left[ Q_\phi(s,a) \right] \right] + \mathbb{E}_{s,a\sim\mathcal{D}} \left[ (Q(s,a) - r(a))^2 \right]$$

**SVR.** The objectives of SVR are given by,

$$\max_{\pi_\theta} \; \mathbb{E}_{s\sim\mathcal{D},\, a\sim\pi_\theta} \left[ Q_\phi(s,a) \right]$$

$$\min_{Q_\phi} \; \alpha \, \mathbb{E}_{s\sim\mathcal{D}} \left[ \mathbb{E}_{a\sim\zeta} \left[ (Q(s,a) - Q_{\min})^2 \right] - \mathbb{E}_{a\sim\mathcal{D}} \left[ \frac{\zeta(a|s)}{\hat{\beta}_\psi(a|s)} (Q(s,a) - Q_{\min})^2 \right] \right]$$

$$+ \mathbb{E}_{s,a\sim\mathcal{D}} \left[ (Q(s,a) - r(a))^2 \right],$$

where $Q_{\min} = \min_{\mathcal{D}} r(a)/(1-\gamma)$, and we set $\zeta$ to the Gaussian model with the same mean as $\pi_\theta$ and a larger variance, as recommended in SVR (Mao et al., 2023).

**Network architecture.** We use a [256, 256, 256]-sized MLP for all neural networks. For all methods, we use the critic network $Q_\phi$ that takes a state-action pair $(s_0, a)$ as input and outputs a scalar estimate of the true action value. For FAC and FQL, we use the behavior proxy policy $\hat{\beta}_\psi$ that takes the dummy state $s_0$, Gaussian noise $z$, and flow time $u \in [0,1]$ as input and then outputs the velocity field $v_u$. The one-step flow policy $\pi_\theta$ that takes the state $s$ and Gaussian noise $z$ and then outputs a single action $a$. For CQL, we use the Gaussian probability policy that takes the dummy state $s_0$. For SVR, we use Gaussian probability models, which takes the dummy state $s_0$ and outputs a action $a$ and its log-density, as the policy and behavior proxy, respectively.

# E    THEORETICAL ANALYSIS

**Proposition 1.** *Let $\beta$ be the underlying behavior policy, $\hat{\beta}$ be our proxy for $\beta$, $\pi$ be the learned actor, and $Q$ be the value function of $\pi$. Consider the original Bellman operator $\mathcal{T}^\pi Q(s,a) = r(s,a) + \gamma \mathbb{E}_{s' \sim P(\cdot|s,a),\, a' \sim \pi(\cdot|s')}\left[Q(s',a')\right]$. Then, in the tabular setting without function approximation, the objective (9) yields the following operator:*

$$\mathcal{T}_{FAC}^\pi Q(s,a) = \begin{cases} \mathcal{T}^\pi Q(s,a) & \text{if } \hat{\beta}(a|s) \geq \epsilon,\ \beta(a|s) > 0 \\ \mathcal{T}^\pi Q(s,a) - \frac{\alpha}{2}\left(\frac{w^{\hat{\beta}}(s,a)\pi(a|s)}{\beta(a|s)}\right) & \text{if } \hat{\beta}(a|s) < \epsilon,\ \beta(a|s) > 0 \\ -\infty & \text{if } \beta(a|s) = 0. \end{cases} \tag{10}$$

*unless $\beta(a|s) = 0$ and $w^{\hat{\beta}}(s,a) = 0$ simultaneously.*

*Proof.* In the tabular setting, we can replace $a \sim \mathcal{D}$ with $a \sim \beta$. Then, the objective (9) can be rewritten as

$$\alpha\, \mathbb{E}_{s \sim \mathcal{D},\, a \sim \pi}\left[w^{\hat{\beta}}(s,a) \cdot Q(s,a)\right] + \mathbb{E}_{s \sim \mathcal{D},\, a \sim \beta}\left[(Q(s,a) - \mathcal{T}^\pi Q(s,a))^2\right]$$

$$= \mathbb{E}_{s \sim \mathcal{D},\, a \sim \beta}\left[\alpha\, w^{\hat{\beta}}(s,a)\frac{\pi(a|s)}{\beta(a|s)} Q(s,a) + (Q(s,a) - \mathcal{T}^\pi Q(s,a))^2\right] \tag{14}$$

We set the derivative of (14) to zero:

$$\alpha\frac{w^{\hat{\beta}}(s,a)\pi(a|s)}{\beta(a|s)} + 2\left(Q(s,a) - \mathcal{T}^\pi Q(s,a)\right) = 0,$$

leading to the solution $Q^*(s,a)$, given by

$$Q^*(s,a) = \mathcal{T}_{FAC}^\pi Q(s,a) = \mathcal{T}^\pi Q(s,a) - \frac{\alpha}{2}\left(\frac{w^{\hat{\beta}}(s,a)\pi(a|s)}{\beta(a|s)}\right).$$

Now consider three cases:

*i)* When $\hat{\beta}(a|s) \geq \epsilon$ and $\beta(a|s) > 0$, $w^{\hat{\beta}}(s,a) = 0$. So, $\mathcal{T}_{FAC}^\pi Q(s,a) = \mathcal{T}^\pi Q(s,a)$.

*ii)* When $\hat{\beta}(a|s) < \epsilon$ and $\beta(a|s) > 0$, $w^{\hat{\beta}}(s,a) > 0$. So, $\mathcal{T}_{FAC}^\pi Q(s,a) = \mathcal{T}^\pi Q(s,a) - \frac{\alpha}{2}\left(\frac{w^{\hat{\beta}}(s,a)\pi(a|s)}{\beta(a|s)}\right)$.

*iii)* When $\beta(a|s) = 0$, the ratio $\pi(a|s)/\beta(a|s) = \infty$ for any $a$ with $w^{\hat{\beta}}(s,a)\pi(a|s) > 0$. Then, its negative becomes $-\infty$.

The only case requiring special care is that $\pi(a|s) > 0$ and $\beta(a|s) = 0$ and $w^{\hat{\beta}}(s,a) = 0$ simultaneously. The ratio $w^{\hat{\beta}}(s,a)/\beta(a|s)$ may cancel out for $\beta(a|s) = 0$ and $w^{\hat{\beta}}(s,a) = 0$. But, it is highly unlikely that we have $\beta(a|s) = 0$ and $w^{\hat{\beta}}(s,a) = 0$ simultaneously by the definition of the weight $w^{\hat{\beta}}(s,a)$. $\square$

With the operator $\mathcal{T}_{FAC}^\pi$ derived above, we now show that the policy (actor) $\pi$ induced by the policy improvement and policy evaluation of FAC is a support-constrained policy, and that the $Q$-function (critic) under the policy $\pi$ yields unbiased $Q$-value estimates in high confidence regions of $\hat{\beta} \geq \epsilon$, while providing under-estimated $Q$-values in low confidence regions of $\hat{\beta} < \epsilon$.

We first define the support-constrained policy, which plays a central role in our analysis, following Kumar et al. (2019); Wu et al. (2022); Mao et al. (2023),

**Definition 1.** *The support-constrained policy class $\Pi_\beta$ is defined as*

$$\Pi_\beta = \{\pi | \pi(a|s) = 0 \ \text{whenever} \ \beta(a|s) = 0\},$$

*where $\beta$ is the underlying behavior policy of the dataset.*

**Proposition 2.** *For any support-constrained policy $\pi$, on the support of $\beta$, $\mathcal{T}_{FAC}^{\pi}$ is a $\gamma$-contraction operator in the $\mathcal{L}_{\infty}$ norm.*

*Proof.* By Proposition 1, we already have

$$\mathcal{T}_{FAC}^{\pi}Q(s,a) = \mathcal{T}^{\pi}Q(s,a) - \frac{\alpha}{2}\left(\frac{w^{\hat{\beta}}(s,a)\pi(a|s)}{\beta(a|s)}\right).$$

Let $Q_1$ and $Q_2$ be two arbitrary functions.

For any $(s,a)$ s.t. $\beta(a|s) > 0$ and $\hat{\beta}(a|s) \geq \epsilon$, i.e., $w^{\hat{\beta}}(s,a) = 0$, we have

$$\begin{aligned}
|\mathcal{T}_{FAC}^{\pi}Q_1(s,a) - \mathcal{T}_{FAC}^{\pi}Q_2(s,a)| &= |\mathcal{T}^{\pi}Q_1(s,a) - \mathcal{T}^{\pi}Q_2(s,a)| \\
&= |\gamma\mathbb{E}_{s'\sim P(\cdot|s,a),a'\sim\pi(\cdot|s')}\left[Q_1(s',a') - Q_2(s',a')\right]| \\
&\leq \gamma\mathbb{E}_{s'\sim P(\cdot|s,a),a'\sim\pi(\cdot|s')}\left[|Q_1(s',a') - Q_2(s',a')|\right] \\
&\leq \gamma\|Q_1 - Q_2\|_{\infty}
\end{aligned}$$

For any $(s,a)$ s.t. $\beta(a|s) > 0$ and $\hat{\beta}(a|s) < \epsilon$, i.e., $w^{\hat{\beta}}(s,a) > 0$, we have

$$\begin{aligned}
|\mathcal{T}_{FAC}^{\pi}Q_1(s,a) - \mathcal{T}_{FAC}^{\pi}Q_2(s,a)| &= |\mathcal{T}^{\pi}Q_1(s,a) - \frac{\alpha}{2}\left(\frac{w^{\hat{\beta}}(s,a)\pi(a|s)}{\beta(a|s)}\right) \\
&\quad - \mathcal{T}^{\pi}Q_2(s,a) + \frac{\alpha}{2}\left(\frac{w^{\hat{\beta}}(s,a)\pi(a|s)}{\beta(a|s)}\right)| \\
&= |\mathcal{T}^{\pi}Q_1(s,a) - \mathcal{T}^{\pi}Q_2(s,a)| \\
&\leq \gamma\|Q_1 - Q_2\|_{\infty}
\end{aligned}$$

For any $(s,a)$ s.t. $\beta(a|s) = 0$, by Definition 1, $\pi(a|s) = 0$, meaning that we can consider only the support of the underlying behavior policy $\beta$.

Therefore, on the support of $\beta$, $\|\mathcal{T}_{FAC}^{\pi}Q_1(s,a) - \mathcal{T}_{FAC}^{\pi}Q_2(s,a)\|_{\infty} \leq \gamma|Q_1 - Q_2\|_{\infty}$, where $\gamma \in [0,1)$.

$\square$

**Theorem 1.** *For any support-constrained policy $\pi$, the fixed point $\bar{Q}^{\pi}$ of $\mathcal{T}_{FAC}^{\pi}$ is*

$$\bar{Q}^{\pi}(s,a) = Q^{\pi}(s,a) - \frac{\alpha}{2}\frac{d^{\pi}(s,a)}{(1-\gamma)}\left(\frac{w^{\hat{\beta}}(s,a)\pi(a|s)}{\beta(a|s)}\right) \quad \text{on the support of } \beta,$$

*where $Q^{\pi}(s,a)$ is the action value function under the policy $\pi$ defined in Section 2.1, and $d^{\pi}(s,a)$ is the normalized discounted state-action distribution (occupancy measure) induced by $\pi$. We further have the following:*

*(i) The fixed point $\bar{Q}^{\pi}$ provides unbiased Q-values for high confidence actions with $\hat{\beta}(a|s) \geq \epsilon$, i.e., $w^{\hat{\beta}}(s,a) = 0$.*

*(ii) The fixed point $\bar{Q}^{\pi}$ provides under-estimated (conservative) Q-values for low confidence actions with $\hat{\beta}(a|s) < \epsilon$, i.e., $w^{\hat{\beta}}(s,a) > 0$.*

*Proof.* By Proposition 2, $\mathcal{T}_{FAC}^{\pi}$ converges to a unique fixed point. That is, for any support-constrained policy $\pi$, the fixed point satisfies $\bar{Q}^{\pi} = \mathcal{T}_{FAC}^{\pi}\bar{Q}^{\pi}$. Furthermore, as in Proposition 2, we can consider only the support of $\beta$.

To represent $\mathcal{T}_{FAC}^{\pi}$ in vector form, as in prior works (Kumar et al., 2020; Mao et al., 2023), we define $R$ as the vector of reward function, and $P^{\pi}(s,a,s',a') := P(s'|s,a)\pi(a'|s')$ as the transition matrix on the state-action pair induced by $\pi$. Then, the fixed point equation is written as follows:

Case (i): For any $(s, a)$ s.t. $\hat{\beta}(a|s) \geq \epsilon$, i.e., $w^{\hat{\beta}}(s, a) = 0$,

$$\bar{Q}^{\pi}(s, a) = \mathcal{T}_{\text{FAC}}^{\pi} \bar{Q}^{\pi}(s, a) \stackrel{(a)}{=} \mathcal{T}^{\pi} \bar{Q}^{\pi}(s, a) = \left[ R + \gamma P^{\pi} \bar{Q}^{\pi} \right](s, a),$$
$$\Rightarrow \bar{Q}^{\pi}(s, a) = \left[ (I - \gamma P^{\pi})^{-1} R \right](s, a) = Q^{\pi}(s, a)$$

where (a) holds due to Proposition 1 in the main paper. Thus, the fixed point provides unbiased $Q$-values for $(s, a)$ s.t. $\hat{\beta}(a|s) \geq \epsilon$.

Case (ii): For any $(s, a)$ s.t. $\hat{\beta}(a|s) < \epsilon$, i.e., $w^{\hat{\beta}}(s, a) \neq 0$,

$$\bar{Q}^{\pi}(s, a) = \mathcal{T}_{\text{FAC}}^{\pi} \bar{Q}^{\pi}(s, a) \stackrel{(a)}{=} \mathcal{T}^{\pi} \bar{Q}^{\pi}(s, a) - \frac{\alpha}{2} \left( \frac{w^{\hat{\beta}}(s, a)\pi(a|s)}{\beta(a|s)} \right),$$

$$= \left[ R + \gamma P^{\pi} \bar{Q}^{\pi} \right](s, a) - \frac{\alpha}{2} \left( \frac{w^{\hat{\beta}}(s, a)\pi(a|s)}{\beta(a|s)} \right)$$

$$\Rightarrow \bar{Q}^{\pi}(s, a) = (I - \gamma P^{\pi})^{-1} \left[ R - \frac{\alpha}{2} \left( \frac{w^{\hat{\beta}}\pi}{\beta} \right) \right](s, a)$$

$$\Rightarrow \bar{Q}^{\pi}(s, a) = Q^{\pi}(s, a) - \frac{\alpha}{2} \left[ (I - \gamma P^{\pi})^{-1} \left( \frac{w^{\hat{\beta}}\pi}{\beta} \right) \right](s, a)$$

where (a) holds due to Proposition 1 in the main paper. The matrix $(I - \gamma P^{\pi})^{-1}$ is the unnormalized discounted state-action distribution induced by the policy $\pi$, i.e., $d^{\pi}(s, a)/(1 - \gamma)$ (refer to Lemma 1.6 of Agarwal et al. (2019)). So, we have the following:

$$\bar{Q}^{\pi}(s, a) = Q^{\pi}(s, a) - \frac{\alpha \, d^{\pi}(s, a)}{2 \, (1 - \gamma)} \left( \frac{w^{\hat{\beta}}(s, a)\pi(a|s)}{\beta(a|s)} \right) \leq Q^{\pi}(s, a)$$

Thus, the fixed point provides under-estimated conservative $Q$-values for $(s, a)$ s.t. $\hat{\beta}(a|s) < \epsilon$. $\square$

Now we want to show that a policy $\pi$ induced by FAC is a support-constrained policy to apply Theorem 1. For this we make the following assumption.

**Assumption 1.** *The well-trained flow behavior proxy in FAC exhibits a support narrower than the support of the underlying true behavior policy, i.e., $supp(\hat{\beta}) \subseteq supp(\beta)$*

In practice, because the dataset has finite cardinality, i.e., $|\mathcal{D}| < \infty$, the support of the dataset is narrower than that of the underlying behavior policy $\beta$, i.e., $supp(\mathcal{D}) \subseteq supp(\beta)$. Since a flow behavior proxy trained on such a dataset inevitably fits to this narrow support, Assumption 1 is valid.

**Theorem 2.** *Under Assumption 1, the policy $\pi$ induced by Eqs. (9) and (11) in FAC is a support-constrained policy. Therefore, the learned $Q$-function under the policy $\pi$, $\bar{Q}^{\pi}$, has unbiased $Q$-values for all $(s, a)$ within high confidence region, i.e., $\hat{\beta}(a|s) \geq \epsilon$, and under-estimated $Q$-values for all $(s, a)$ within low confidence region, i.e., $\hat{\beta}(a|s) < \epsilon$.*

*Proof.* The objective (11) of policy improvement in FAC has the following relation (Park et al., 2025):

$$\mathbb{E}_{s \sim \mathcal{D}, a \sim \pi} \left[ \bar{Q}^{\pi}(s, a) \right] - \lambda \mathbb{E}_{s \sim \mathcal{D}, z \sim \mathcal{N}(0, I)} \left[ \| a_{\theta}(s, z) - a_{\psi}(s, z) \|_2^2 \right]$$
$$\leq \mathbb{E}_{s \sim \mathcal{D}, a \sim \pi} \left[ \bar{Q}^{\pi}(s, a) \right] - \lambda \mathbb{E}_{s \sim \mathcal{D}} \left[ W_2(\pi, \hat{\beta})^2 \right], \tag{15}$$

where $\bar{Q}^{\pi}$ is the learned $Q$-function under the policy $\pi$, and $W_2(\pi, \hat{\beta})^2$ is the squared 2-Wasserstein distance between $\pi$ and $\hat{\beta}$.

For ease of exposition, we convert the unconstrained optimization problem (15) to the following constrained one, for some $\delta \geq 0$,

$$\max_{\pi} \mathbb{E}_{s \sim \mathcal{D}, a \sim \pi} \left[ \bar{Q}^{\pi}(s, a) \right] \tag{16}$$
$$\text{s.t. } W_2(\pi, \hat{\beta})^2 < \delta.$$

This constrained optimization problem for policy improvement leads to the following observations:

1. The constraint of the problem (16) offers no incentive for the policy $\pi$ to allocate probability density on actions outside the support of the behavior policy $\beta$, as ensured by the Assumption 1.

2. The objective of the problem (16), which yields a greedy policy whose $\bar{Q}^\pi$ value becomes $-\infty$ outside the support of the underlying behavior policy, according to Proposition 1, derives the policy $\pi$ to assign zero probability density to actions outside the support of the behavior policy $\beta$. If $\pi$ assigns a positive density to those actions, i.e., $\pi(a|s) > 0$, then the objective becomes

$$\mathbb{E}_{s\sim\mathcal{D},\, a\sim\pi}\left[\bar{Q}^\pi(s,a)\right] = \int_{s\in\mathcal{D}, a\in\mathcal{A}} \pi(a|s)\bar{Q}^\pi(s,a)dads = -\infty.$$

Thus, the greedy policy $\pi$ have to assign zero probability density to actions outside the support of the underlying behavior policy $\beta$.

Based on these observations, we conclude that the policy $\pi$ learned by FAC is a support-constrained policy, i.e., $\pi \in \Pi_\beta$.

Since the policy $\pi$ of FAC is a support-constrained policy, by Theorem 1, the learned $\bar{Q}^\pi$ has a unique fixed point and provides unbiased $Q$-values for all $(s,a)$ within high confidence region, i.e., $\hat{\beta}(a|s) \geq \epsilon$, and under-estimated $Q$-values for all $(s,a)$ within low confidence region, i.e., $\hat{\beta}(a|s) < \epsilon$. $\qquad\square$

# F   ADDITIONAL RESULTS

## F.1   OFFLINE RL RESULTS

Table 3 presents evaluation results on the 55 singletask variants in the OGBench. For each task category, we select the best hyperparameters on its default task and evaluate the remaining tasks with the same ones. For pixel-based tasks, we reuse the hyperparameters selected for their state-based tasks; for example, `visual-cube-single-play-singletask-task1-v0` is evaluated using the hyperparameters selected for `cube-single-play-singletask-v0`. The detailed experimental settings are provided in Appendix G.3.

Table 3: Full offline RL evaluation results on the OGBench. (∗) indicates the default task in each task category. We report the final performance averaged over 8 seeds (4 seeds for pixel-based tasks), with ± indicating the standard deviation.

| Task | Gaussian Policies | | | Diffusion Policies | | | Flow Policies | | | | |
|---|---|---|---|---|---|---|---|---|---|---|---|
| | BC | IQL | ReBRAC | IDQL | SRPO | CAC | FAWAC | FBRAC | IFQL | FQL | FAC (Ours) |
| antmaze-large-navigate-singletask-task1-v0 (∗) | 0 | 48 | 91 | 0 | 0 | 42 | 1 | 70 | 24 | 80 | **94.0**$_{\pm3.0}$ |
| antmaze-large-navigate-singletask-task2-v0 | 6 | 42 | **88** | 14 | 4 | 1 | 0 | 35 | 8 | 57 | 86.0$_{\pm5.7}$ |
| antmaze-large-navigate-singletask-task3-v0 | 29 | 72 | 51 | 26 | 3 | 49 | 12 | 83 | 52 | 93 | **97.5**$_{\pm2.1}$ |
| antmaze-large-navigate-singletask-task4-v0 | 8 | 51 | 84 | 62 | 45 | 17 | 10 | 37 | 18 | 80 | **89.5**$_{\pm7.7}$ |
| antmaze-large-navigate-singletask-task5-v0 | 10 | 54 | 90 | 2 | 1 | 55 | 9 | 76 | 38 | 83 | **96.0**$_{\pm4.8}$ |
| antmaze-giant-navigate-singletask-task1-v0 (∗) | 0 | 0 | **27** | 0 | 0 | 0 | 0 | 0 | 0 | 4 | 6.5$_{\pm5.2}$ |
| antmaze-giant-navigate-singletask-task2-v0 | 0 | 1 | 16 | 0 | 0 | 0 | 0 | 4 | 0 | 9 | **37.5**$_{\pm15.0}$ |
| antmaze-giant-navigate-singletask-task3-v0 | 0 | 0 | **34** | 0 | 0 | 0 | 0 | 0 | 0 | 0 | 0.5$_{\pm1.4}$ |
| antmaze-giant-navigate-singletask-task4-v0 | 0 | 0 | 5 | 0 | 0 | 0 | 0 | 9 | 0 | 14 | **20.0**$_{\pm17.9}$ |
| antmaze-giant-navigate-singletask-task5-v0 | 1 | 19 | 49 | 0 | 0 | 0 | 0 | 6 | 13 | 16 | **50.5**$_{\pm29.4}$ |
| humanoidmaze-medium-navigate-singletask-task1-v0 (∗) | 1 | 32 | 16 | 1 | 0 | 38 | 6 | 25 | 69 | 19 | **71.5**$_{\pm14.7}$ |
| humanoidmaze-medium-navigate-singletask-task2-v0 | 1 | 41 | 18 | 1 | 1 | 47 | 40 | 76 | 85 | **94** | 88.0$_{\pm12.1}$ |
| humanoidmaze-medium-navigate-singletask-task3-v0 | 6 | 25 | 36 | 0 | 2 | 83 | 19 | 27 | 49 | 74 | **95.5**$_{\pm3.3}$ |
| humanoidmaze-medium-navigate-singletask-task4-v0 | 0 | 0 | 15 | 1 | 1 | 5 | 1 | 1 | 1 | 3 | **25.0**$_{\pm9.0}$ |
| humanoidmaze-medium-navigate-singletask-task5-v0 | 2 | 66 | 24 | 1 | 3 | 91 | 31 | 63 | **98** | 97 | **98.0**$_{\pm2.1}$ |
| humanoidmaze-large-navigate-singletask-task1-v0 (∗) | 0 | 3 | 2 | 0 | 0 | 1 | 0 | 0 | 6 | 7 | **15.0**$_{\pm19.8}$ |
| humanoidmaze-large-navigate-singletask-task2-v0 | 0 | 0 | 0 | 0 | 0 | 0 | 0 | 0 | 0 | 0 | 0.0$_{\pm0.0}$ |
| humanoidmaze-large-navigate-singletask-task3-v0 | 1 | 7 | 8 | 3 | 1 | 2 | 1 | 10 | **48** | 11 | 20.5$_{\pm18.4}$ |
| humanoidmaze-large-navigate-singletask-task4-v0 | 1 | 1 | 1 | 0 | 0 | 0 | 0 | 0 | 1 | 2 | **4.5**$_{\pm11.2}$ |
| humanoidmaze-large-navigate-singletask-task5-v0 | 0 | 1 | **2** | 0 | 0 | 0 | 0 | 1 | 0 | 1 | 1.5$_{\pm2.1}$ |
| antsoccer-arena-navigate-singletask-task1-v0 | 2 | 14 | 0 | 44 | 2 | 1 | 22 | 17 | 61 | 77 | **82.0**$_{\pm4.8}$ |
| antsoccer-arena-navigate-singletask-task2-v0 | 2 | 17 | 0 | 15 | 3 | 0 | 8 | 8 | 75 | 88 | **93.5**$_{\pm4.2}$ |
| antsoccer-arena-navigate-singletask-task3-v0 | 0 | 6 | 0 | 0 | 0 | 8 | 11 | 16 | 14 | 61 | **62.5**$_{\pm11.5}$ |
| antsoccer-arena-navigate-singletask-task4-v0 (∗) | 1 | 3 | 0 | 0 | 0 | 0 | 12 | 24 | 16 | 39 | **53.0**$_{\pm5.6}$ |
| antsoccer-arena-navigate-singletask-task5-v0 | 0 | 2 | 0 | 0 | 0 | 0 | 9 | 15 | 0 | 36 | **47.5**$_{\pm15.3}$ |
| cube-single-play-singletask-task1-v0 | 10 | 88 | 89 | 95 | 89 | 77 | 81 | 73 | 79 | 97 | **99.0**$_{\pm1.9}$ |
| cube-single-play-singletask-task2-v0 (∗) | 3 | 85 | 92 | 96 | 82 | 80 | 81 | 83 | 73 | 97 | **100.0**$_{\pm0.0}$ |
| cube-single-play-singletask-task3-v0 | 9 | 91 | 93 | 99 | 96 | 98 | 87 | 82 | 88 | 98 | **100.0**$_{\pm0.0}$ |
| cube-single-play-singletask-task4-v0 | 2 | 73 | 92 | 93 | 70 | 91 | 79 | 79 | 79 | 94 | **98.5**$_{\pm3.0}$ |
| cube-single-play-singletask-task5-v0 | 3 | 78 | 87 | 90 | 61 | 80 | 78 | 76 | 77 | 93 | **96.5**$_{\pm3.3}$ |
| cube-double-play-singletask-task1-v0 | 8 | 27 | 45 | 39 | 7 | 21 | 21 | 47 | 35 | **61** | 60.0$_{\pm11.3}$ |
| cube-double-play-singletask-task2-v0 (∗) | 0 | 1 | 7 | 16 | 0 | 2 | 2 | 22 | 9 | 36 | **37.5**$_{\pm10.0}$ |
| cube-double-play-singletask-task3-v0 | 0 | 0 | 4 | 17 | 0 | 3 | 1 | 4 | 8 | 22 | **31.5**$_{\pm10.8}$ |
| cube-double-play-singletask-task4-v0 | 0 | 0 | 1 | 0 | 0 | 0 | 0 | 0 | 1 | **5** | 4.0$_{\pm3.7}$ |
| cube-double-play-singletask-task5-v0 | 0 | 4 | 4 | 1 | 0 | 3 | 2 | 2 | 17 | 19 | **32.5**$_{\pm6.2}$ |
| scene-play-singletask-task1-v0 | 19 | 94 | 95 | **100** | 94 | **100** | 87 | 96 | 98 | **100** | **100.0**$_{\pm0.0}$ |
| scene-play-singletask-task2-v0 (∗) | 1 | 12 | 50 | 33 | 2 | 50 | 18 | 46 | 0 | 76 | **100.0**$_{\pm0.0}$ |
| scene-play-singletask-task3-v0 | 1 | 32 | 55 | 94 | 4 | 49 | 38 | 78 | 54 | **98** | 97.0$_{\pm2.8}$ |
| scene-play-singletask-task4-v0 | 2 | 0 | 3 | 4 | 0 | 0 | 6 | 4 | 0 | 5 | **58.0**$_{\pm38.4}$ |
| scene-play-singletask-task5-v0 | 0 | 0 | 0 | 0 | 0 | 0 | 0 | 0 | 0 | 0 | **1.5**$_{\pm4.2}$ |
| puzzle-3x3-play-singletask-task1-v0 | 5 | 33 | 97 | 52 | 89 | 97 | 25 | 63 | 94 | 90 | **100.0**$_{\pm0.0}$ |
| puzzle-3x3-play-singletask-task2-v0 | 1 | 4 | 1 | 0 | 0 | 0 | 4 | 2 | 1 | 16 | **100.0**$_{\pm0.0}$ |
| puzzle-3x3-play-singletask-task3-v0 | 1 | 3 | 3 | 0 | 0 | 0 | 1 | 1 | 0 | 10 | **100.0**$_{\pm0.0}$ |
| puzzle-3x3-play-singletask-task4-v0 (∗) | 1 | 2 | 2 | 0 | 0 | 0 | 1 | 2 | 0 | 16 | **100.0**$_{\pm0.0}$ |
| puzzle-3x3-play-singletask-task5-v0 | 1 | 3 | 5 | 0 | 0 | 0 | 1 | 2 | 0 | 16 | **100.0**$_{\pm0.0}$ |
| puzzle-4x4-play-singletask-task1-v0 | 1 | 12 | 26 | 48 | 24 | 44 | 1 | 32 | 49 | 34 | **52.0**$_{\pm25.1}$ |
| puzzle-4x4-play-singletask-task2-v0 | 0 | 7 | 12 | 14 | 0 | 0 | 0 | 5 | 4 | **16** | 7.5$_{\pm7.5}$ |
| puzzle-4x4-play-singletask-task3-v0 | 0 | 9 | 15 | 34 | 21 | 29 | 1 | 20 | 50 | 18 | **62.0**$_{\pm13.4}$ |
| puzzle-4x4-play-singletask-task4-v0 (∗) | 0 | 5 | 10 | 26 | 7 | 1 | 0 | 5 | 21 | 11 | **35.0**$_{\pm12.4}$ |
| puzzle-4x4-play-singletask-task5-v0 | 0 | 4 | 7 | **24** | 1 | 0 | 0 | 4 | 2 | 7 | 5.0$_{\pm5.1}$ |
| visual-cube-single-play-singletask-task1-v0 | - | 70.0 | **83.0** | - | - | - | - | 55.0 | 49.0 | 81.0 | 82.0$_{\pm5.2}$ |
| visual-cube-double-play-singletask-task1-v0 | - | 34.0 | 4.0 | - | - | - | - | 6.0 | 8.0 | 21.0 | **48.0**$_{\pm7.3}$ |
| visual-scene-play-singletask-task1-v0 | - | 97.0 | 98.0 | - | - | - | - | 46.0 | 86.0 | 98.0 | **99.0**$_{\pm2.0}$ |
| visual-puzzle-3x3-play-singletask-task1-v0 | - | 7.0 | 88.0 | - | - | - | - | 7.0 | **100.0** | 94.0 | 95.0$_{\pm2.0}$ |
| visual-puzzle-4x4-play-singletask-task1-v0 | - | 0.0 | 26.0 | - | - | - | - | 0.0 | 8.0 | 33.0 | **56.0**$_{\pm11.3}$ |

### F.1.1 COMPARISON WITH SEQUENCE MODEL BASED OFFLINE RL METHODS

We compare FAC against offline RL methods using sequence model architectures: Transformer (Vaswani et al., 2017) and state space model (Gu et al., 2022). Unlike Multi Layer Perceptron (MLP) based policies, sequence model based policies can condition on extended state-action-reward histories. For this comparison, we report results in the original papers and compare against Transformer-based methods Decision Transformer (DT) (Chen et al., 2021), DC (Kim et al., 2024), QDT (Yamagata et al., 2023), Reinformer (Zhuang et al., 2024), and state space model based method Decision Mamba (DM) Lv et al. (2024).

On the D4RL MuJoCo tasks, FAC outperforms all sequence model based methods in terms of overall score, despite without explicit history conditioning. On the D4RL Antmaze tasks, FAC also outperforms Transformer-based methods. Notably, Reinformer exhibits strong performance on the simpler umaze tasks but degrades sharply in the more challenging medium tasks, whereas FAC maintains high performance across medium and large tasks.

Table 4: Comparison with sequence model based baselines. For MuJoCo datasets, we use the following abbreviations: *m* for *medium*, *mr* for *medium-replay*, *me* for *medium-expert*.

| MuJoCo Tasks | Transformer based | | | | SSM based | MLP based |
| | DT | DC | QDT | Reinformer | DM | FAC (Ours) |
| --- | --- | --- | --- | --- | --- | --- |
| halfcheetah-m | 42.6 | 43.0 | 42.3 | 42.9 | 43.8 | $65.0_{\pm1.5}$ |
| hopper-m | 67.6 | 92.5 | 66.5 | 81.6 | 98.5 | $91.9_{\pm3.9}$ |
| walker2d-m | 74.0 | 79.2 | 67.1 | 80.5 | 80.3 | $85.2_{\pm0.9}$ |
| halfcheetah-mr | 36.6 | 41.3 | 35.6 | 39.0 | 40.8 | $55.4_{\pm2.7}$ |
| hopper-mr | 82.7 | 94.2 | 52.1 | 83.3 | 89.1 | $99.1_{\pm0.9}$ |
| walker2d-mr | 66.6 | 76.6 | 58.2 | 72.9 | 79.3 | $83.0_{\pm5.8}$ |
| halfcheetah-me | 86.8 | 93.0 | - | 92.0 | 93.5 | $101.9_{\pm5.6}$ |
| hopper-me | 107.6 | 110.4 | - | 107.8 | 111.9 | $104.2_{\pm4.6}$ |
| walker2d-me | 108.1 | 109.6 | - | 109.4 | 111.6 | $108.4_{\pm0.6}$ |
| **Average** | 74.7 | 82.2 | - | 78.8 | 83.2 | 88.2 |

| Antmaze Tasks | Transformer based | | | | SSM based | MLP based |
| | DT | DC | QDT | Reinformer | DM | FAC (Ours) |
| --- | --- | --- | --- | --- | --- | --- |
| umaze | - | 85.0 | - | 84.4 | 100.0 | $98.5_{\pm3.0}$ |
| umaze-diverse | - | 78.5 | - | 65.8 | 90.0 | $93.5_{\pm6.0}$ |
| medium-play | - | - | - | 13.2 | - | $88.0_{\pm9.6}$ |
| medium-diverse | - | - | - | 10.6 | - | $85.0_{\pm7.3}$ |
| large-play | - | - | - | - | - | $90.0_{\pm4.3}$ |
| large-diverse | - | - | - | - | - | $88.0_{\pm6.0}$ |
| **Average** | - | - | - | - | - | 90.5 |

## F.2 OFFLINE-TO-ONLINE RL RESULTS

Figure 7 shows the offline-to-online results on the 5 default singletask variants in the OGBench and 6 tasks in the D4RL Antmaze and 4 tasks in the D4RL Adroit. In all result plots, the first $1M$ gradient steps denote the offline training phase (the gray shaded region), and the online finetuning phase starts after $1M$ steps. For each tasks, FAC reuses the same hyperparameters used in the offline RL evaluation, whereas FQL uses hyperparameters specified for offline-to-online evaluation in its original paper (Park et al., 2025). The detailed experimental settings for the offline-to-online evaluation are provided in Appendix G.3.3.

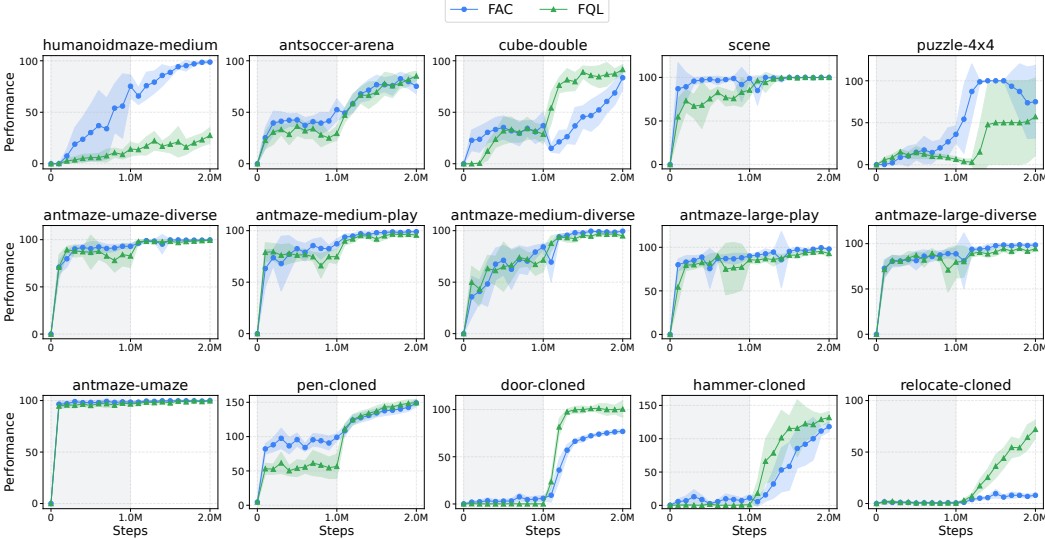

Figure 7: Offline-to-online results on 15 tasks, averaged over 8 seeds with standard deviation. The gray shaded region indicates the offline phase, and the online fine-tuning phase is left unshaded.

# G    EXPERIMENTAL DETAILS

We implement the proposed algorithm in JAX (Bradbury et al., 2018) on top of an official implementation of Park et al. (2025), with reference to Lipman et al. (2024).

## G.1    BENCHMARKS

We evaluate our proposed method on 50 state-based and 5 pixel-based singletasks of OGBench (Park et al., 2024c) and 23 tasks of D4RL (Fu et al., 2020).

**OGBench.** We evaluate FAC across the following tasks from 10 navigation and manipulation environments:

- `antmaze-large-navigate-singletask-task{1,2,3,4,5}-v0`
- `antmaze-giant-navigate-singletask-task{1,2,3,4,5}-v0`
- `humanoidmaze-medium-navigate-singletask-task{1,2,3,4,5}-v0`
- `humanoidmaze-large-navigate-singletask-task{1,2,3,4,5}-v0`
- `antsoccer-arena-navigate-singletask-task{1,2,3,4,5}-v0`
- `cube-single-play-singletask-task{1,2,3,4,5}-v0`
- `cube-double-play-singletask-task{1,2,3,4,5}-v0`
- `scene-play-singletask-task{1,2,3,4,5}-v0`
- `puzzle-3x3-play-singletask-task{1,2,3,4,5}-v0`
- `puzzle-4x4-play-singletask-task{1,2,3,4,5}-v0`
- `visual-cube-single-play-singletask-task1-v0`
- `visual-cube-double-play-singletask-task1-v0`
- `visual-scene-play-singletask-task1-v0`
- `visual-puzzle-3x3-play-singletask-task1-v0`
- `visual-puzzle-4x4-play-singletask-task1-v0`

OGBench provides a suite of environments and datasets for offline goal-conditioned RL. We use datasets from 5 navigation environments (`antmaze-large`, `antmaze-giant`, `humanoidmaze-medium`, `humanoidmaze-large`, `antsoccer-arena`) and 5 manipulation environments (`cube-single`, `cube-double`, `scene`, `puzzle-3x3`, `puzzle-4x4`). Each dataset offers 5 singletask variants. For offline RL evaluation, we use the reward-based singletask variants whose rewards are relabeled to align with the singletask specification. In navigation tasks, a sparse reward function is used, the agent receives a reward of 0 on successfully reaching the goal position and -1 otherwise. In manipulation tasks, each singletask is composed of multiple subtasks, and the agent receives the negative of the number of unsuccessful subtasks as its reward.

The navigation environments include `antmaze` and `humanoidmaze`, where a quadrupedal agent and a humanoid one are required to reach goal positions in a given maze, and `antsoccer`, where a quadrupedal agent dribbles a ball to a goal position. The manipulation environments include `cube`, where a robot arm is required to pick and place colored cubes, and `scene`, where a robot arm executes a sequence of subtasks, and `puzzle`, where a robot arm solves a lights-out puzzle.

**D4RL.** We evaluate FAC across the following tasks from the MuJoCo, Antmaze, Adroit domains:

- `halfcheetah-medium-v2`
- `hopper-medium-v2`
- `walker2d-medium-v2`
- `halfcheetah-medium-replay-v2`
- `hopper-medium-replay-v2`
- `walker2d-medium-replay-v2`
- `halfcheetah-medium-expert-v2`
- `hopper-medium-expert-v2`

- `walker2d-medium-expert-v2`
- `antmaze-umaze-v2`
- `antmaze-umaze-diverse-v2`
- `antmaze-medium-play-v2`
- `antmaze-medium-diverse-v2`
- `antmaze-large-play-v2`
- `antmaze-large-diverse-v2`
- `pen-human-v1`
- `pen-cloned-v1`
- `door-human-v1`
- `door-cloned-v1`
- `hammer-human-v1`
- `hammer-cloned-v1`
- `relocate-human-v1`
- `relocate-cloned-v1`

D4RL provides standard offline RL datasets for locomotion (MuJoCo domain), navigation (Antmaze domain), and dexterous manipulation (Adroit domain). In the MuJoCo domain, we evaluate on `halfcheetah`, `hopper`, `walker2d` with three datasets `medium`, `medium-replay`, `medium-expert`. `medium` dataset consists of transition rollouts from a partially trained behavior policy, and `medium-replay` dataset is the full replay buffer of the behavior policy, and `medium-expert` dataset mixes trajectories from the medium and expert-level behavior policies. In the Antmaze domain, we use {`umaze`, `medium`, `large`}-sized mazes with two datasets `play` and `diverse`. The two datasets contain transitions to reach from start positions to goal positions. In the Adroit domain, we use `pen`, `door`, `hammer`, `relocate` tasks with two datasets `human` and `cloned`. The `human` dataset consists of tele-operated demonstrations, and the `cloned` dataset contains transition rollouts from a behavior-cloned policy trained on the `human` dataset.

## G.2 BASELINES

We primarily present the official reported performance from each baseline paper for benchmark datasets. For datasets that are not reported in the baseline papers, we refer to performances from other papers reporting performance for those datasets. For datasets without reported performances from other papers, we reproduced the results using an official implementation of baselines, with tuning hyperparameters.

Specifically, all results for our method and the reproduced baselines are reported as the mean $\pm$ standard deviation over 8 random seeds.

**For OGBench,** we obtain the reported performances of BC, IQL, ReBRAC, IDQL, SRPO, CAC, FAWAC, FBRAC, IFQL, FQL from Park et al. (2025).

**For D4RL-MuJoCo domain,** we obtain the reported performance of TD3+BC, CQL from Kostrikov et al. (2021). Additionally, we reproduce FQL and tune a broader range of hyperparameters than the ones recommended in the original paper. The hyperparameters and training settings for FQL are summarized in Table 5 & 6. In particular, we consider normalizing $Q$-values in the actor objective, following TD3+BC (Fujimoto & Gu, 2021). Empirically, enabling or disabling this $Q$-normalization exhibits almost similar performance. To maintain consistency with default hyperparameter settings of FQL, we therefore report results obtained without the $Q$-normalization in the actor objective.

**For D4RL-Antmaze domain,** we obtain the reported performance of TD3+BC from Kostrikov et al. (2021), the reported performance of MCQ from Yeom et al. (2024), ones of SAC-RND from (Tarasov et al., 2023), and the reported performance of CAC from the original paper and Park et al. (2025).

**For D4RL-Adroit domain,** we obtain the reported performance of TD3+BC, IQL, SAC-RND from Tarasov et al. (2023) and the reported performance of IDQL, SRPO, CAC from Park et al. (2025).

Table 5: Hyperparameters for FQL on the D4RL-MuJoCo domain.

| Hyperparameters | Value |
| --- | --- |
| Learning rate | 0.0003 |
| Optimizer | Adam (Kingma & Ba, 2014) |
| Gradient Steps | 1000000 |
| Minibatch size | 256 |
| MLP dimension | [512, 512, 512, 512] |
| Nonlinearity | GELU (Hendrycks & Gimpel, 2016) |
| Target network smoothing coefficient | 0.005 |
| Discount factor $\gamma$ | 0.99 |
| Flow steps | 10 |
| Flow time sampling distribution | $\text{Unif}([0, 1])$ |
| Clipped double Q-learning | True |
| BC coefficient $\lambda$ | Table 6 |
| Q-normalization in actor loss | False |

Table 6: BC Coefficient $\lambda$ for FQL on the D4RL-MuJoCo domain.

| Tasks (Datasets) | BC coefficient $\lambda$ |
| --- | --- |
| halfcheetah-medium-v2 | 3.0 |
| hopper-medium-v2 | 100.0 |
| walker2d-medium-v2 | 300.0 |
| halfcheetah-medium-replay-v2 | 30.0 |
| hopper-medium-replay-v2 | 100.0 |
| walker2d-medium-replay-v2 | 300.0 |
| halfcheetah-medium-expert-v2 | 30.0 |
| hopper-medium-expert-v2 | 100.0 |
| walker2d-medium-expert-v2 | 300.0 |

## G.3 OUR ALGORITHM

### G.3.1 IMPLEMENTATION DETAILS

We describe the implementation details for flow actor-critic (FAC).

**Network architecture.** We use a [512, 512, 512, 512]-sized MLP for all neural networks. We apply layer normalization (Ba et al., 2016) to critic networks. The critic network $Q_\phi$ takes a state-action pair $(s, a)$ as input and outputs a scalar estimate of the expected return under the one-step flow policy $\pi_\theta$. The behavior proxy policy $\hat{\beta}_\psi$ takes state $s$, Gaussian noise $z \in \mathbb{R}^{d_\mathcal{A}}$, and flow time $u \in [0, 1]$ as inputs and outputs the velocity field $v_u$. The one-step flow policy $\pi_\theta$ takes state $s$ and Gaussian noise $z$ and then outputs a single action $a_\theta(s, z)$.

**Image processing.** For pixel-based tasks, following Park et al. (2025), we use a small variant of the IMPALA encoder Espeholt et al. (2018) and apply a random shift augmentation with probability of $0.5$ and use frame stacking with three image observations.

**Flow matching proxy.** We use the flow matching objective (6) (Lipman et al., 2022; Park et al., 2025). We use the Euler method with 10 steps to sample actions $a_\psi(s, z)$ and evaluate behavior proxy density $\hat{\beta}_\psi(a|s)$.

Specifically, the proxy density can be evaluated exactly via the Instantaneous Change of Variables (Chen et al., 2018), which requires the divergence of the velocity field $\nabla_{a_u} \cdot v_u(a_u; s, u)$. This is the trace of a Jacobian with respect to $a_u$, and it scales as $\mathcal{O}((d_\mathcal{A})^2)$ (Grathwohl et al., 2018), which may be prohibitive for high-dimensional action spaces. To reduce computational cost without introducing bias, we can approximate the divergence using Hutchinson's unbiased estimator (Hutchinson, 1989; Adams et al., 2018), lowering the complexity to $\mathcal{O}(d_\mathcal{A})$. In practice, we adopt this estimator for OGBench-Humanoidmaze and D4RL-Adroit domains, which have action space dimension $d_\mathcal{A} > 8$, and using the estimator exhibits performance comparable to exact evaluation.

**Critic learning.** We use twin critic networks to improve stability (Fujimoto et al., 2018). For the empirical Bellman operator in the critic loss, we can aggregate them either by the arithmetic mean of the two $Q$ values or by their minimization. We present the aggregation choices in Table 7.

**One-step flow policy learning.** Following (Fujimoto & Gu, 2021), we apply the $Q$-value normalization in the one-step flow actor loss to balance the $Q$-maximizing term and the distance regularization term, i.e., $\max_{\pi_\theta} \mathbb{E}_{a\sim\pi_\theta}\left[\frac{Q_\phi(s,a)}{|Q_\phi|}\right] + \lambda D(\pi_\theta, \hat{\beta}_\psi)$, where $|Q_\phi| = \frac{1}{M}\sum_m |Q_\phi(s_m,a)|$ with mini-batch $\{s_m\}_{m=1}^M$ and actions $a \sim \pi_\theta(\cdot|s_m)$. In practice, this normalization reduces the sensitivity to the actor regularization coefficient $\lambda$ and enables a common tuning range across offline RL tasks.

**Training and Evaluation.** We train FAC for 1M gradient steps on all the state-based tasks of OGBench and D4RL, and for 500K gradient steps on the pixel-based tasks of OGBench. We evaluate it every 100K updates using 25 episodes. At evaluation, the actor samples exactly one action at each time step. One thing to note is that we report the last performance measured after 1M gradient steps.

### G.3.2 HYPERPARAMETERS

All hyperparameters used for the main results in Table 1 & 2 are summarized in Table 7 & 8. The critic penalization coefficient $\alpha$ and actor regularization coefficient $\lambda$ are important in our method. The entire set of the two hyperparameters $(\alpha,\ \lambda)$ used across all tasks is $\alpha \in \{0.05, 0.1, 0.5, 1.0, 5.0, 10.0\}$ and $\lambda \in \{0.0003, 0.001, 0.003, 0.1, 0.3, 1.0, 3.0, 10.0\}$. For task-specific hyperparameters and a narrow range for tuning hyperparameters, please refer to Table 8.

Another important hyperparameter is the type of dataset-driven threshold $\epsilon$ in the weight $w^{\hat{\beta}_\psi}(s,a)$. As described, for datasets where the state coverage is broad but the action diversity is limited, we use the dataset-wide constant threshold (OGBench: Antmaze, Antsoccer tasks / D4RL: Antmaze tasks). For all remaining tasks, we use the batch-adaptive threshold.

Other hyperparameter is the use of clipped double Q-learning. With twin critics, the empirical Bellman target can be computed using either the minimization or the mean of the two $Q$ values for state-action pairs. We use the minimization on the D4RL tasks, and the mean on OGBench tasks other than Antmaze tasks. This selection aligns with the standard configurations of the baseline methods.

Table 7: Hyperparameters for our method

| Hyperparameters | Value |
|---|---|
| Learning rate | 0.0003 |
| Optimizer | Adam (Kingma & Ba, 2014) |
| Gradient Steps | 1M (default), 500K (`OGBench:pixel-based tasks`) |
| Minibatch size | 256 |
| Epochs for flow proxy model | 250 (default), |
| | 125 (`D4RL:MuJoCo, OGBench:pixel-based tasks`) |
| MLP dimension | [512, 512, 512, 512] |
| Nonlinearity | GELU (Hendrycks & Gimpel, 2016) |
| Target network smoothing coefficient $\rho$ | 0.005 |
| Discount factor $\gamma$ | 0.995 |
| Step count of the Euler method | 10 |
| Flow time sampling distribution | Unif($[0,1]$) |
| Q-normalization in actor loss | True |
| Clipped double Q-learning | Min (`D4RL`: default / `OGBench:Antmaze`) |
| | Mean (`D4RL`: Null / `OGBench`: default) |
| Estimator for evaluating $\hat{\beta}_\psi(a\|s)$ | Exact (`D4RL`: default / `OGBench`: default) |
| | Hutchinson's estimator with 8 probes |
| | (`D4RL:Adroit` / `OGBench:Humanoidmaze`) |
| Dataset-driven threshold $\epsilon$ | Batch-adaptive (`D4RL`: default / `OGBench`: default) |
| | Dataset-wide constant |
| | (`D4RL:Antmaze` / `OGBench:Antmaze,Antsoccer`) |
| Critic penalization coefficient $\alpha$ | Table 8 |
| Actor regularization coefficient $\lambda$ | Table 8 |

Table 8: Critic penalization coefficient $\alpha$ and actor regularization coefficient $\lambda$ of FAC

(a) OGBench

| Tasks (Datasets) | $\alpha$ | $\lambda$ |
|---|---|---|
| antmaze-large-navigate-singletask-task{1,2,3,4,5}-v0 | 0.5 | 0.1 |
| antmaze-giant-navigate-singletask-task{1,2,3,4,5}-v0 | 1.0 | 0.1 |
| humanoidmaze-medium-navigate-singletask-task{1,2,3,4,5}-v0 | 0.5 | 0.3 |
| humanoidmaze-large-navigate-singletask-task{1,2,3,4,5}-v0 | 0.5 | 0.1 |
| antsoccer-arena-navigate-singletask-task{1,2,3,4,5}-v0 | 1.0 | 0.1 |
| {cube, visual-cube}-single-play-singletask-task{1,2,3,4,5}-v0 | 0.5 | 10.0 |
| {cube, visual-cube}-double-play-singletask-task{1,2,3,4,5}-v0 | 1.0 | 1.0 |
| {scene, visual-scene}-play-singletask-task{1,2,3,4,5}-v0 | 1.0 | 1.0 |
| {puzzle, visual-puzzle}-3x3-play-singletask-task{1,2,3,4,5}-v0 | 1.0 | 1.0 |
| {puzzle, visual-puzzle}-4x4-play-singletask-task{1,2,3,4,5}-v0 | 5.0 | 0.3 |

(b) D4RL: MuJoCo domain

| Tasks (Datasets) | $\alpha$ | $\lambda$ |
|---|---|---|
| halfcheetah-medium-v2 | 0.05 | 0.0003 |
| hopper-medium-v2 | 5.0 | 0.1 |
| walker2d-medium-v2 | 5.0 | 0.03 |
| halfcheetah-medium-replay-v2 | 0.05 | 0.0003 |
| hopper-medium-replay-v2 | 5.0 | 0.1 |
| walker2d-medium-replay-v2 | 5.0 | 0.1 |
| halfcheetah-medium-expert-v2 | 0.5 | 0.003 |
| hopper-medium-expert-v2 | 5.0 | 0.3 |
| walker2d-medium-expert-v2 | 5.0 | 0.03 |

(c) D4RL: Antmaze domain

| Tasks (Datasets) | $\alpha$ | $\lambda$ |
|---|---|---|
| antmaze-umaze-v2 | 1.0 | 0.1 |
| antmaze-umaze-diverse-v2 | 1.0 | 0.1 |
| antmaze-medium-play-v2 | 0.5 | 0.03 |
| antmaze-medium-diverse-v2 | 5.0 | 0.1 |
| antmaze-large-play-v2 | 5.0 | 0.03 |
| antmaze-large-diverse-v2 | 1.0 | 0.03 |

(d) D4RL: Adroit domain

| Tasks (Datasets) | $\alpha$ | $\lambda$ |
|---|---|---|
| pen-human-v1 | 10.0 | 0.3 |
| pen-cloned-v1 | 1.0 | 0.1 |
| door-human-v1 | 1.0 | 3.0 |
| door-cloned-v1 | 5.0 | 10.0 |
| hammer-human-v1 | 0.5 | 0.03 |
| hammer-cloned-v1 | 0.5 | 1.0 |
| relocate-human-v1 | 5.0 | 10.0 |
| relocate-cloned-v1 | 5.0 | 10.0 |

### G.3.3 DETAILS FOR ONLINE FINE-TUNING

FAC can extend to the online finetuning setting (Ball et al., 2023; Nakamoto et al., 2023; Shin et al., 2025; 2026; Kostrikov et al., 2021; Nair et al., 2020). For the offline-to-online evaluation, we initialize the one-step flow actor, critics, and flow behavior proxy using the networks trained for

$1M$ gradient steps in the offline training, and then continue training for an additional $1M$ steps with online interactions. We evaluate it every 100K updates. The practical implementation of FAC for online finetuning is provided in Algorithm 2.

No additional hyperparameters are introduced for the online finetuning phase, and we reuse the value of the hyperparameters used for offline training. The only modifications for the online finetuning are: (i) jointly training the flow behavior proxy together with the flow actor and critics, and (ii) replacing the dataset wide constant $\min_{(s,a)\sim\mathcal{D}}\hat{\beta}_\psi(a|s)$ with a batch wide constant $\min_{(s,a)\sim\mathcal{B}}\hat{\beta}_\psi(a|s)$, where $\mathcal{B}$ is a mini-batch for each gradient step. During online fine-tuning, the flow behavior proxy is trained so that it can represent online samples $(s, a, r, s')$ and provide density estimates. The dataset wide constant requires estimating densities for the entire samples in the replay buffer at each gradient steps, even though only a smaller size of mini-batch is used for each steps. This leads to unnecessary computation during online finetuning. To avoid this, while preserving the minimization-based thresholding design, we adopt the batch wide constant, which estimates the minimization of densities only for the mini-batch samples. We note that the batch adaptive threshold is used without modification during the online finetuning.

---

**Algorithm 2:** Flow Actor-Critic for online finetuning

---

offline-trained one-step flow policy network $\pi_\theta$, offline-trained critic networks $Q_{\phi_1}, Q_{\phi_2}$, offline-trained target networks $Q_{\bar{\phi}_1}, Q_{\bar{\phi}_2}$, and offline-trained flow behavior proxy network $\hat{\beta}_\psi$

**for** *each iteration* **do**

    Sample $(s, a, r, s') \sim \mathcal{D}$, $z \sim \mathcal{N}(0, I)$, $u \sim \text{Unif}([0, 1])$ and compute $\epsilon$ by eq. (7).

    Update $\hat{\beta}_\psi$ with eq. (6).

    Update $Q_{\phi_i}$ with eq. (9), $\quad i \in \{1, 2\}$.

    Update $\pi_\theta$ with eq. (11).

    $\bar{\phi}_i \leftarrow \rho\phi_i + (1 - \rho)\bar{\phi}_i, \quad i \in \{1, 2\}$ for some $\rho$.

---

## H    COMPUTATIONAL COSTS

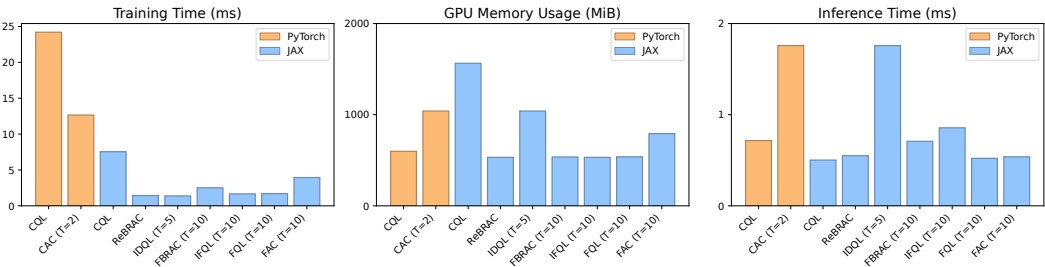

Figure 8: Computational costs of FAC and baselines

We compare the computational costs of FAC against various baselines in terms of training time, GPU memory usage, and inference time. For all baselines, we primarily use their official open-source implementations whenever available. For ReBRAC, we use an open-source implementation provided with FQL Park et al. (2025), and FBRAC is newly implemented. For the diffusion-based baseline CAC, its official implementation is based on Pytorch Paszke et al. (2019). To enable a fair comparison, we additionally report the Pytorch-based CQL implementation and the JAX-based one from the official open-source implementation of Cal-ql Nakamoto et al. (2023).

The comparisons are conducted on a single RTX 3090 GPU. Training time is measured as the wall-clock time for one gradient step with a batch size of 256, and inference time is measured as the time required to produce a single action sample. The computational costs are shown in Figure 8. For diffusion-based and flow-based methods, we denote the number of denoising or flow steps as a suffix to the method name (e.g., FAC (T=10)).

For training time, the PyTorch-based baselines exhibit larger training time than JAX-based methods. Within the JAX-based methods, IDQL shows the lowest training time. The training time of FAC is moderately higher than IDQL and the multi-step flow policy baselines, reflecting the additional cost of density estimation. Nevertheless, FAC requires faster training time than JAX-based CQL, even though both methods employ conservative critic objectives. The difference comes from simple practical implementation of FAC. The practical implementation of CQL requires auxiliary action sampling and additional gradient computing for soft maximization over Q-values, whereas FAC requires sampling only a single action without additional computation.

For GPU memory usage, JAX-based CQL and the diffusion-based baselines consume substantially higher GPU memory than other methods.

For inference time, diffusion-based baselines, CAC (T=2) and IDQL (T=5), require higher time than other baselines due to iterative denoising or diffusion steps. In contrast, the one-step flow actor used by FAC and FQL yields inference time comparable to Gaussian policy based baselines.

