# OpenReview forum: "Flow Actor-Critic for Offline Reinforcement Learning"
_ICLR.cc/2026/Conference — ICLR 2026 Poster_

### Official Review · Reviewer_FJQh · 2025-10-27

**Soundness:** 2
**Presentation:** 3
**Contribution:** 3
**Rating:** 6
**Confidence:** 3

**Summary:**

This paper proposes a Flow Actor-Critic, a new actor-critic method for offline RL, based on recent flow policies.

**Strengths:**

This paper proposes a new critic regularizer to achieve accurate Q-value estimation under OOD conditions, and provides extensive experimental validation. The paper is overall well-organized.

**Weaknesses:**

1. In the Introduction, the paper mentions that diffusion policies make policy optimization computationally heavy. How does the proposed method's computational complexity compare to that of diffusion-based policies? It is suggested that the author compares the hardware efficiency of the proposed method with existing methods using metrics such as wall-clock throughput, FLOPs, and VRAM usage.
2. In the Introduction, it is unclear why the highly expressive flow behavior model and accompanying density can solve the OOD problem.
3. The paper lacks theoretical proof for why the proposed method provides more accurate Q-value estimation under OOD conditions compared to existing methods.
4. The authors compare many baselines, but fewer methods from 2024 and 2025 are included. It is suggested to compare with more recent methods published in the last two years.
5. From Table 2 in the main text, the proposed method's advantages are not evident, as in many cases, its performance is worse than the baseline. More experimental results showing the advantages of the proposed method should be added or moved from the appendix to the main text.
6. It is suggested to include Q estimate results in the main text, comparing the Q-values of the proposed method with the baseline methods and the true Q-value to demonstrate the superiority of the proposed method.

**Questions:**

1. In the Introduction, the paper mentions that diffusion policies make policy optimization computationally heavy. How does the proposed method's computational complexity compare to that of diffusion-based policies? It is suggested that the author compares the hardware efficiency of the proposed method with existing methods using metrics such as wall-clock throughput, FLOPs, and VRAM usage.
2. In the Introduction, it is unclear why the highly expressive flow behavior model and accompanying density can solve the OOD problem.
3. The paper lacks theoretical proof for why the proposed method provides more accurate Q-value estimation under OOD conditions compared to existing methods.
4. The authors compare many baselines, but fewer methods from 2024 and 2025 are included. It is suggested to compare with more recent methods published in the last two years.
5. From Table 2 in the main text, the proposed method's advantages are not evident, as in many cases, its performance is worse than the baseline. More experimental results showing the advantages of the proposed method should be added or moved from the appendix to the main text.
6. It is suggested to include Q estimate results in the main text, comparing the Q-values of the proposed method with the baseline methods and the true Q-value to demonstrate the superiority of the proposed method.

---

> ### Author Response · Authors · 2025-11-23
> **Authors' Response to Reviewer FJQh**
>
> We thank the reviewer for valuable comments and constructive suggestion.
> Please note that the numbers of equations, figures, tables and sections   below refer to those in the revised paper available at openreview unless we give special remarks.
> Our responses to the comments are presented below.
>
> ### R1. Computational cost (Question 1)
>
> We compared the computational costs of FAC against various baselines in terms of training time, GPU memory usage, and inference time, and the result is shown in Figure 8 of Appendix H, presenting the cost measurements for CQL(PyTorch), CAC(PyTorch), CQL(JAX), ReBRAC(JAX), IDQL(JAX), FBRAC(JAX), IFQL(JAX), FQL(JAX), and FAC(JAX).
>
> Training time: Among all baselines, the PyTorch implementations of CQL and CAC exhibit the largest training time. Within the JAX-based methods, IDQL shows the lowest training time. The training time of FAC is moderately higher than IDQL and the multi-step flow policy baselines, reflecting the additional cost of density estimation. Nevertheless, FAC requires faster training time than CQL(JAX), even though both methods employ conservative critic objectives.
>
> GPU memory usage: CQL(JAX) and the diffusion-based policies consume substantially higher GPU memory than other methods.
>
> Inference time: For action sampling, CAC(with T=2) and IDQL(with T=5) require the highest inference cost due to iterative denoising. In contrast, flow policy based methods(with T=10) require inference times comparable to Gaussian policy based method.
>
> We appreciate the reviewer's constructive comments regarding the computational burden of diffusion policies mentioned in the Introduction. Based on this feedback, we have revised the Introduction in the revised version to accurately reflect the empirical results.
>
> ### R2. Unclear explanation in the introduction (Question 2)
>
> We thank the reviewer for pointing out the lack of clarity in the Introduction. We have revised the manuscript and added an explicit explanation in the Introduction to clarify how the expressiveness of the flow behavior model and its density estimation contribute to addressing the OOD issue.
>
>
> ### R3. Theoretical proof (Question 3)
>
> As described in Proposition 1 of Sec. 4.1, our critic learning yields the following operator: $\mathcal{T}^{\pi}_{FAC}Q(s,a)=\mathcal{T}^{\pi}Q(s,a)-\frac{\alpha}{2}\left(w^{\hat{\beta}}(s,a)\pi(a|s)/\beta(a|s)\right)$, where $\mathcal{T}^{\pi}Q(s,a)$ is the standard Bellman operator. Consider a dataset where expert-level samples are relatively sparse compared to medium-level samples. In such dataset, the underlying behavior density $\beta(a|s)$ may be low for expert-level samples, which would increase the penalizing term in the FAC operator. However, if the weight $w^{\hat{\beta}}(s,a)$ correctly identifies these samples as confident ID samples, the penalizing term becomes zero, enabling the critic to learn more accurate Q-values for these expert-level samples.
>
> In contrast, as in the CQL paper [1], its operator $\mathcal{T}^{\pi}_{CQL}Q(s,a)=\mathcal{T}^{\pi}Q(s,a)-\alpha\left((\pi(a|s)/\beta(a|s))-1\right)$ applies strong penalization to low density samples. For expert-level samples with low density $\beta(a|s)$, CQL impose a large penalizing, pushing their Q-values below their true Q-values. This can hinder the actor from sampling expert-level actions, since the actor aims to maximize its learned Q-values.
>
> Therefore, the proposed critic learning in FAC preserves high-value expert samples, while penalizing genuinely OOD samples.
>
> [1] Kumar, Aviral, et al. "Conservative q-learning for offline reinforcement learning." Advances in neural information processing systems 33 (2020): 1179-1191.
>
> ### R4. Baselines (Question 4)
>
> We have listed all baselines published in or after 2024 that are relavant to our setting.
> - Gaussian-based: EPQ (NeurIPS 2024)
> - Diffusion-based: SRPO (ICLR 2024), CAC (ICLR 2024)
> - Flow-based: QIPO (ICLR 2025), FBRAC\&FAWAC\&IFQL\&FQL (ICML 2025)
>
> The remaining baselines are also strong baselines referred in many offline RL works.
>
> It would be great if the reviewer can further suggest  additional methods that should be compared against. We will include them in the references and, if necessary, add their results to the performance tables.

---

> ### Author Response · Authors · 2025-11-23
> **Authors' Response to Reviewer FJQh**
>
> ### R5. FAC on the D4RL tasks (Question 5)
>
> As discussed in Sec. 6.3, FAC demonstrates clear advantages over diffusion- and flow-based baselines across all D4RL tasks. FAC consistently achieves higher overall performance than CAC, IDQL, SRPO, QIPO-OT, FQL, which indicates that the proposed density-based critic penalization provides beyond the capabilities of existing diffusion- and flow-based methods.
>
> Furthermore, on the MuJoCo and Adroit tasks, FAC achieves performance comparable to the strongest Gaussian policy based baselines such as ReBRAC. Importantly, on the more challenging OGBench tasks, FAC achieves about twice overall performance of ReBRAC.
>
> If there are specific cases where the reviewer finds FAC's advantages insufficiently emphasized, we welcome the feedback and will incorporate the additions in the paper.
>
> ### R6. Q-bias (Question 6)
>
> We thank the reviewer for the insightful suggestion. We agree on that comparing Q-value estimates against the true Q-values would provide an informative perspective on Q-bias. We tried to compute the true Q-values in the offline RL tasks. However, this requires repeatedly resetting the environments to identical states. We have continued exploring whether such an evaluation can be made feasible, and we aim to have results within the rebuttal.
>
> We would like to highlight the example presented in Figure 5 in Appendix C, which visualizes the Q-value estimates and sampled action histograms for AR (a variant of FQL) and AR+CP (a variant of FAC), along with the ground truth values and the OOD regions. Although this example is based on a simple setting, it might clearly illustrate how the proposed critic penalization shapes the Q-function and subsequently guides the flow actor. We believe this example provides a intuitive demonstration of how FAC reduces overestimation in low-density regions.
>
> Thank you again for your time and effort.

---

> > ### Comment · Reviewer_FJQh · 2025-11-28
> >
> > Thank you for your response. I still have some concerns:
> > 1. Regarding the theoretical proof, the paper compares the operator of CQL, showing that its Q-value is lower than the true value, but it does not analyze the operators of other typical DRL algorithms. Is the claim that the proposed method provides more accurate Q-value estimates than existing DRL methods a general conclusion? Additionally, regarding the claim that the proposed method offers more accurate Q-value estimates under OOD conditions, I suggest a more rigorous mathematical derivation to support this, such as deriving the difference between the Q-value learned by the proposed method and the true Q-value, as well as the difference between existing DRL methods and the true Q-value, to demonstrate that the proposed method has a smaller difference from the true Q-value.
> > 2. Regarding the baseline, for example, CAC (ICLR 2024), the paper cites CAC (Ding & Jin, 2023), which may cause confusion. I recommend updating the citation. Additionally, regarding offline DRL baselines, the current experiment covers a limited range of baselines. Some other types of baselines, such as algorithms from the Decision Transformer series, could also be included for comparison.
> > 3. In Table 2 of the main text, for example, in the MuJoCo tasks like HalfCheetah-m, Hopper-m, and others, the proposed FAC (Ours) seems not to outperform the baseline in some tasks.
> > 4. It is necessary to show a comparison of the Q-values between the proposed method and the baseline, as this would provide a clearer visual indication of how the proposed method's Q-value is closer to the true Q-value compared to the baseline methods.

---

> ### Author Response · Authors · 2025-12-03
>
> We thank the reviewer for further discussion. Our response to the feedback is given below.
>
> ### Theoretical Analysis and Evaluation and Visudal Indication of Quality of Learned Q function
>
> We added a rigorous theoretical analysis of the policy (actor) and Q-function (critic) learned by FAC in Appendix J. Proposition 2 and Theorems 1 and 2 in Appendix J show that the Q-function learned from the FAC critic update (eq.9 in Sec. 4.1) has a unique fixed point that yields true  Q values on state-action pairs with high confidence $\hat{\beta}(a|s)\geq\epsilon$ and under-estimated conservative Q values on the state-action pairs with low confidence $\hat{\beta}(a|s)<\epsilon$. In addition, the policy induced by the FAC actor update is shown to be a support-constrained policy, $\pi$ such that $\pi(a|s)=0$ whenever $\beta(a|s)=0$. Please see Appendix J for details.
>
>
> Regarding the accuracy assessment of the learned Q function and  comparison with baselines:  We now provided a concrete example in Appendix C of the revised paper, replacing the old example in Appendix C.
> Since it is difficult to know the true action value function in most RL tasks, we considered a continuous action bandit problem, which is modified from the bandit example in Chapter 2 of Sutton and Barto, 2018. As seen if Figure 5 of Appendix C, when the true action value function has asymmetric two modes and the offline dataset has large supoptimal samples with small near-optimal samples, the learned action value functions of existing methods: CQL, SVR and FQL definitely show degradation such as failing to suppress OOD regions or too much underestimation even in the ID region (in CQL). On the other hand, FAC successfully learns the Q function. Please see Appendix C of the revised paper for details.
>
>
>
>
>
>
> ### Comparison with more baselines including sequence-based methods
>
> We  have updated the citations for SRPO, CAC, FQL, and QIPO-OT. Please see Sec. 6.1 in the revised paper.
>
> We further compared FAC against offline RL methods using sequence model architectures: Transformer and state space model. We compared against Transformer-based methods Decision Transformer (DT; NeurIPS 21), DC (ICLR 24), QDT (ICML 23), Reinformer (ICML 24), and state space model based method Decision Mamba (DM; NeurIPS 24). Their performance was taken from their original papers. This comparison result is provided  in Table 8 in Appendix I. We observe that FAC yields superior performance. The corresponding results are provided in the following tables for convenience
>
> |MuJoCo Tasks|DT|DC|QDT|Reinformer|DM|FAC (Ours)|
> |-|-|-|-|-|-|-|
> |halfcheetah-m|42.6|43.0|42.3|42.9|43.8|65.0±1.5|
> |hopper-m|67.6|92.5|66.5|81.6|98.5|91.9±3.9|
> |walker2d-m|74.0|79.2|67.1|80.5|80.3|85.2±0.9|
> |halfcheetah-mr|36.6|41.3|35.6|39.0|40.8|55.4±2.7|
> |hopper-mr|82.7|94.2|52.1|83.3|89.1|99.1±0.9|
> |walker2d-mr|66.6|76.6|58.2|72.9|79.3|83.0±5.8|
> |halfcheetah-me|86.8|93.0|-|92.0|93.5|101.9±5.6|
> |hopper-me|107.6|110.4|-|107.8|111.9|104.2±4.6|
> |walker2d-me|108.1|109.6|-|109.4|111.6|108.4±0.6|
> |Average|74.7|82.2|-|78.8|83.2|88.2|
>
> |Antmaze Tasks|DT|DC|QDT|Reinformer|DM|FAC (Ours)|
> |-|-|-|-|-|-|-|
> |umaze|-|85.0|-|84.4|100.0|98.5±3.0|
> |umaze-diverse|-|78.5|-|65.8|90.0|93.5±6.0|
> |medium-play|-|-|-|13.2|-|88.0±9.6|
> |medium-diverse|-|-|-|10.6|-|85.0±7.3|
> |large-play|-|-|-|-|-|90.0±4.3|
> |large-diverse|-|-|-|-|-|88.0±6.0|
> |Average|-|-|-|-|-|90.5|

---

### Official Review · Reviewer_Ypn5 · 2025-10-31

**Soundness:** 4
**Presentation:** 3
**Contribution:** 3
**Rating:** 8
**Confidence:** 4

**Summary:**

The paper introduces Flow Actor-Critic (FAC) for offline reinforcement learning. The key idea is to pair a flow-based behavior proxy with a one-step flow actor and to use the proxy’s density to tell when an action looks in-distribution versus out-of-distribution. The critic is then penalized only in regions where the behavior density is low, reducing the usual overestimation on OOD actions without dampening values where the data actually supports them. The actor, meanwhile, is trained to maximize Q while staying close to the behavior proxy, which helps with stability and makes good use of the expressiveness of flows without resorting to costly multi-step sampling. Empirically, FAC delivers strong results across large offline RL suites (OGBench and D4RL), beating or matching strong Gaussian, diffusion, and prior flow baselines. The ablations support the main claims: when candidate action sets get larger (which typically increases OOD risk), FAC remains robust while variants without the density-aware critic degrade.

**Strengths:**

The paper is clearly written and well structured. The motivation is crisp, the method is introduced in a logical sequence, and the experiments are laid out so the key results are easy to grasp before diving into details.

The core idea is clean and elegant. By using the behavior model’s density to decide when to be conservative, the critic only gets pushed down where actions look out-of-distribution and is left alone where the data provide strong support. Coupling this with a one-step flow actor keeps the approach expressive yet practical to optimize.

The empirical results are very strong. FAC matches or beats strong Gaussian, diffusion, and prior flow baselines across broad offline RL suites (OGBench and D4RL). The improvements are consistent rather than cherry-picked, and the gains on OGBench are clearly significant.

The ablation study is very insightful as well. "Effect of the critic penalization" section clearly shows why the density-aware critic matters. Variants without it degrade as OOD risk increases, while sweeps over key hyperparameters demonstrate robustness.

**Weaknesses:**

I don’t see any major weaknesses. The only area that feels under-explored is the threshold design used to decide when the behavior density is “low.” The paper offers two options—a dataset-wide constant and a batch-adaptive threshold—but stops short of examining how this choice affects robustness across tasks. It would strengthen the work to add a small, focused study on the threshold itself: for example, trying simple variants like scaling the base threshold by a weight; using per-batch quantile thresholds instead of a fixed cutoff; or annealing the threshold during training.

**Questions:**

This method relies on a learned behavior density to decide where to be conservative. What happens if that density is hard to estimate well, for example, the dataset is multi-modal with rare but good actions, or high-dimensional with sparse coverage? In those cases, could FAC wrongly mark good actions as low-density and hurt performance?

---

> ### Author Response · Authors · 2025-11-23
> **Authors' Response to Reviewer Ypn5**
>
> We thank the reviewer for valuable comments and insightful suggestion.  Please note that the numbers of equations, figures, tables and sections   below refer to those in the revised paper available at openreview unless we give special remarks.  Our responses to the comments are presented below.
>
> ### R1. Dataset-driven method for $\epsilon$ (Weakness 1)
>
> The $\epsilon$ in the critic penalization weight $w^{\hat{\beta}}(s,a)$ determines whether a given state-action sample $(s,a)$ should be treated as reliably in-distribution. Since $\hat{\beta}_{\psi}(a|s)$ depends on both the states and the actions taken at those states, the effectiveness of $\epsilon$ depends on the state-action structure of the dataset.
> Motivated by the reviewer's insightful suggestion, we additionally explored a batch wide constant $\epsilon$, whose results are in Figure 3(e) in Sec. 6.4.
> Below, we focus on explaining the intent behind the two original $\epsilon$ design choices based on the empirical observations from offline RL tasks. Figure 3(e) in Sec. 6.4 presents the final performance of FAC under different $\epsilon$ designs, the corresponding results are provided in the following table for convenience:
>
> |Tasks|Batch Adaptive|Dataset wide Constant|
> |-|-:|-:|
> |halfcheetah-medium-expert|101.9±5.6|71.9±4.8|
> |antmaze-medium-play|86.0±2.3|88.0±9.6|
> |antmaze-large-diverse|76.0±8.7|88.0±6.0|
>
>
> **1. Batch adaptive $\epsilon$:**
> Offline dataset often combine trajectories generated by behavior policies with different skill levels. To illustrate an extreme case, consider a dataset collected from medium-level and expert-level behavior policies that visit almost identical states. Although their state coverage is similar, the action distribution at the same state can differ significantly: the expert policy tends to exhibit substantially narrower action support than the medium policy.
>
> This creates a characteristic multi-modal, uneven state-action density: for the same state $s$, the medium-level actions occupy a relatively broad region, while expert-level actions form sharp modes with much higher density. If $\epsilon$ is fixed according to the lower-density component (e.g., medium behavior), the ratio $\hat{\beta}(a|s)/\epsilon$ can easily exceed 1 even for actions that are genuinely OOD around the expert mode. This causes the critic penalization weight to vanish for those OOD actions.
>
> We ran an experiment regarding $\epsilon$ on halfcheetah-medium-expert-v2, a dataset that, we hypothesize, exhibits narrow state coverage and highly uneven action densities. The result in the above table show that batch adaptive $\epsilon$ yields superior performance in this case. Our interpretation is that when the dataset contains multiple action modes per state with large density disparities, adapting $\epsilon$ to each mini-batch sample reflects the local density scale and prevents unintended disappearance of critic penalization.
>
> **2. Dataset wide constant $\epsilon$:**
> Conversely,  batch adaptive $\epsilon$ yields a bit degraded performance in tasks such antmaze-large-diverse, as seen in the above table. Our analysis is that Antmaze datasets differ fundamentally from halfcheetah-medium-expert. The D4RL antmaze dataset has very wide state coverage, however, at each state, the behavior policy generates only a small number of actions, often forming narrow and irregular local densities. This indicates that $\hat{\beta}_{\psi}(a|s)/\epsilon$ might fluctuate excessively under the batch adaptive $\epsilon$. The batch adaptive value of $\epsilon$ can become excessively large, which may penalize even actions that are essential for stitching trajectories from the start position to the goal position.
>
> To mitigate this effect, we introduced the dataset-wide constant $\epsilon$, providing a coherent lower bound on $\hat{\beta}_{\psi}(a|s)$, preventing over-penalization of states whose action samples are sparse and densities are uneven. With dataset-wide constant $\epsilon$, FAC achieves the highest performance on antmaze-large-diverse-v2, as seen in Figure 3(e) in Sec. 6. Furthermore, to avoid excessive hyperparameter tuning for $\epsilon$, we fixed the antmaze domain to use the dataset-wide constant $\epsilon$.
>
> **3. Robust performance across both $\epsilon$ schemes:**
> Outside of the above extremes, we observe that the two $\epsilon$ designs often perform similarly well. For instance, in antmaze-medium-play, both batch adaptive and dataset-wide constant $\epsilon$ achieve comparable performance even under the same critic penalization coefficient $\alpha$ and actor regularization coefficient $\lambda$. We conjecture that this result comes from the intermediate of behavior distribution of the two extreme datasets.

---

> ### Author Response · Authors · 2025-11-23
> **Authors' Response to Reviewer Ypn5**
>
> ### R2. The fidelity of flow matching model (Question 1)
>
> We investigated how the fidelity of the flow behavior proxy affects both sampling quality and density estimation.
> Figure 6 in Appendix D.1 shows the quality of generated samples and density estimation obtained from a single flow matching model with different step counts $T\in\{3,5,10,30,50\}$. Using the same trained flow matching model, we vary only the step counts of the Euler method for sampling and density estimation. When $T=\{3,5\}$, decreasing step counts leads to a clear degradation in both sampling and density estimation, with $T=3$ failing to reliably separate ID and OOD regions. In contrast, for $T\geq 10$, the sampling performance is satisfactory, and density estimates contrast is enough to distinguish ID and OOD regions.
>
> This observation is  because the Euler method servers as the numerical integration solver, and the step counts directly affects the fidelity of the approximated flow. When the learned flow matching model represents almost linear trajectories from base noises $x_0$ to data samples $x_1$,  step count 1 can be sufficient. However, for complex data distributions where the learned flow trajectories exhibit curve, higher curvature needs more integration steps to ensure sufficiently accurate sampling and density estimation.
>
> Based on this observation, we performed further experiment on  the step count of the Euler method in two RL tasks and evaluated the final performance of FAC, shown in Figure 3(c) in Sec. 6.4. The results are summarized in the table below. In humanoidmaze-medium-navigate-task1, performance drops when $T<10$, whereas $T=30$ yields a slight improvement over $T=10$. Similarly, in puzzle-3x3-play-task4, performance steadily decreases as $T$ decreases below 10, whereas $T=30$ achieves the same 100\% success rate as $T=10$. These results demonstrate that $T=10$ provides a favorable trade-off between computational cost and performance.
>
> |Tasks|T=1|T=3|T=5|T=10|T=30|
> |-|-:|-:|-:|-:|-|
> |humanoidmaze-medium-navigate-task1|25.5±23.9|28.0±22.2|30.0±16.4|71.5±14.7|75.0±5.1|
> |puzzle-3x3-play-task4|86.5±21.6|91.5±18.4|96.5±9.9|100.0±0.0|100.0±0.0|
>
> Thank you again for your time and effort.

---

### Official Review · Reviewer_XLd9 · 2025-10-31

**Soundness:** 2
**Presentation:** 3
**Contribution:** 3
**Rating:** 6
**Confidence:** 3

**Summary:**

This paper proposes Flow Actor-Critic (FAC), a new offline reinforcement learning algorithm that leverages flow-based generative models to jointly regularize the actor and penalize the critic. Unlike previous flow policy methods that only used flow models for action sampling, FAC fully exploits the learned flow behavior proxy policy to estimate tractable behavior densities. These densities are then used to (1) identify out-of-distribution (OOD) regions for critic penalization and (2) constrain the actor through a one-step flow mapping toward the behavior support. The authors derive the modified Bellman operator under this framework and provide extensive empirical evaluation on OGBench and D4RL, demonstrating improved or state-of-the-art performance compared to Gaussian- and diffusion-based baselines.

**Strengths:**

Originality: The paper introduces a novel and coherent idea — using the same flow model both for actor regularization and critic penalization. This dual use of flow behavior density provides a principled way to directly detect OOD regions, addressing a key challenge in offline RL.

Technical quality: The method is conceptually sound and the derivation of the flow-based critic operator is clear. The integration of density-weighted Q penalization and flow-regularized actor optimization is well motivated.

Empirical validation: Experiments are extensive, covering 50 OGBench tasks and 23 D4RL tasks, with comparisons to strong baselines such as ReBRAC, IQL, CQL, IDQL, CAC, and FQL. FAC consistently achieves top or near-top results.

Significance: The approach bridges expressive policy modeling (via flows) and conservative value estimation in a unified framework. It represents a meaningful step forward in making expressive policies practically viable for offline RL.

**Weaknesses:**

Incremental over FQL: Conceptually, the work extends FQL by reusing the flow density for critic penalization rather than introducing an entirely new framework. Although the empirical gains are notable, the conceptual advance may be viewed as incremental.

Computational cost: Flow-based models are typically heavier than Gaussian or VAE-based policies, but the paper does not discuss computational trade-offs.

**Questions:**

How sensitive is FAC to the choice of ε and to inaccuracies in the flow-based density estimate? Could an adaptive or uncertainty-aware threshold improve robustness?

How does FAC compare computationally to diffusion-based methods (e.g., CAC, IDQL) and multi-step flow policies (e.g., IFQL)?

Can the authors provide quantitative results showing the quality of their flow density estimates versus ground-truth or surrogate densities?

How does FAC behave on datasets with limited coverage or severe distributional shift, where even flow models may fail to represent $\beta(a|s)$?

Could FAC be extended to online or offline-to-online fine-tuning, where exploration or policy improvement beyond the dataset is needed?

---

> ### Author Response · Authors · 2025-11-23
> **Authors' Response to Reviewer XLd9**
>
> We thank the reviewer for valuable comments.  Our responses to the comments are presented below. Please note that the numbers of equations, figures, tables and sections below refer to those in the revised paper available at openreview unless we give special remarks.
>
> ### R1. Novelty (Weakness 1)
>
> While FAC adopts the same actor-regularization design as FQL, the core contribution of FAC is to exploit density estimation of flow model. In contrast to FQL, which primarily leverages the flow matching model for multi-modal action sampling, FAC introduces a new critic-penalization mechanism that explicitly exploits the density estimation capability of the flow model.
>
> FAC uses the flow behavior proxy (flow matching model) not only for sampling but also for estimating a strong density estimate $\hat{\beta}(a|s)$, which allows us to identify low-density state-action pairs and penalize them for critic learning. This density-based mechanism is not present in FQL.
>
> Thus, FAC introduces a conceptually distinct critic optimization, enabling the flow behavior proxy to play a fundamentally different role beyond action sampling. The empirical improvements are nor merely due to architectural reuse, but come from a new way of integrating flow models in critic learning.
>
>
> ### R2. Computational cost (Weakness 2 \& Question 2)
>
> We compared the computational costs of FAC against various baselines in terms of training time, GPU memory usage, and inference time, and the result is shown in Figure 8 of Appendix H, presenting the cost measurements for CQL(PyTorch), CAC(PyTorch), CQL(JAX), ReBRAC(JAX), IDQL(JAX), FBRAC(JAX), IFQL(JAX), FQL(JAX), and FAC(JAX).
>
> Training time: Among all baselines, the PyTorch implementations of CQL and CAC exhibit the largest training time. Within the JAX-based methods, IDQL shows the lowest training time. The training time of FAC is moderately higher than IDQL and the multi-step flow policy baselines, reflecting the additional cost of density estimation. Nevertheless, FAC requires faster training time than CQL(JAX), even though both methods employ conservative critic objectives.
>
> GPU memory usage: CQL(JAX) and the diffusion-based policies consume substantially higher GPU memory than other methods.
>
> Inference time: For action sampling, CAC(with T=2) and IDQL(with T=5) require the highest inference cost due to iterative denoising. In contrast, flow policy based methods(with T=10) require inference times comparable to Gaussian policy based method.

---

> ### Author Response · Authors · 2025-11-23
> **Authors' Response to Reviewer XLd9**
>
> ### R3. Fidelity of the flow behavior proxy (Question 1\&3)
>
> We investigated how the fidelity of the flow behavior proxy affects both sampling quality and density estimation.
> Figure 6 in Appendix D.1  shows the quality of generated samples and density estimation obtained from a single flow matching model with different step counts $T\in\{3,5,10,30,50\}$, with the colorbar of each log-density heatmap to show quantitative results. Using the same trained flow matching model, we vary only the step counts of the Euler method for sampling and density estimation. When $T=\{3,5\}$, decreasing step counts leads to a clear degradation in both sampling and density estimation, with $T=3$ failing to reliably separate ID and OOD regions. In contrast, for $T\geq 10$, the sampling performance is satisfactory, and density estimates contrast is enough to distinguish ID and OOD regions.
>
> This observation is because the Euler method servers as the numerical integration solver, and the step counts directly affects the fidelity of the approximated flow. When the learned flow matching model represents almost linear trajectories from base noises $x_0$ to data samples $x_1$,  step count  1 can be sufficient. However, for complex data distributions where the learned flow trajectories exhibit curve, higher curvature needs more integration steps to ensure sufficiently accurate sampling and density estimation.
>
> Based on this observation, we further ablate the step count of the Euler method in two RL tasks and evaluate the final performance of FAC, as shown in Figure 3(c) in Sec. 6.4. The results are summarized in the table below. In humanoidmaze-medium-navigate-task1, performance drops when $T<10$, whereas $T=30$ yields a slight improvement over $T=10$. Similarly, in puzzle-3x3-play-task4, performance steadily decreases as $T$ decreases below 10, whereas $T=30$ achieves the same 100\% success rate as $T=10$. These results demonstrate that $T=10$ provides a favorable trade-off between computational cost and performance.
>
> |Tasks|T=1|T=3|T=5|T=10|T=30|
> |-|-:|-:|-:|-:|-|
> |humanoidmaze-medium-navigate-task1|25.5±23.9|28.0±22.2|30.0±16.4|71.5±14.7|75.0±5.1|
> |puzzle-3x3-play-task4|86.5±21.6|91.5±18.4|96.5±9.9|100.0±0.0|100.0±0.0|
>
>
> ### R4. Critic penalization weight (Question 1\&3)
>
> As shown in Figure 6 in Appendix D.1, the log-density of the flow behavior proxies exhibits a scale difference from the ground truth log-density. Nevertheless, $T\geq 10$, the flow matching model provides strong density contrast to distinguish between ID and OOD samples. This observation motivated the design of our critic penalization weight, which leverages relative density difference rather than relying on absolute density scales.
>
> Our key design principle in the weight (eq.8) is to use the ratio between the flow behavior proxy density of the action generated from the flow one-step actor and that of the dataset actions. As shown in Figure 1, the flow matching model can provide strong density contrast to distinguish ID and OOD regions. However, offline RL tasks might contain more diverse state-action pairs, making it difficult to rely on the absolute scale of the estimated densities. In practice, due to the limited dataset, the estimated density might be differ in scale from the true underlying densities on entire state-action space (e.g., being unnormalized). Correcting this mismatch would require estimating densities across the entire state-action space to compute an appropriate normalization constant, which is infeasible. To avoid this issue, we design the weight using the ratio $\hat{\beta}_{\psi}(a|s)/\epsilon$, where  $\epsilon$ is defined by the densities of the dataset samples.

---

> ### Author Response · Authors · 2025-11-23
> **Authors' Response to Reviewer XLd9**
>
> ### R5. Dataset-driven method for $\epsilon$ (Question 1)
>
> The $\epsilon$ in the critic penalization weight $w^{\hat{\beta}}(s,a)$ determines whether a given state-action sample $(s,a)$ should be treated as reliably in-distribution. Since $\hat{\beta}_{\psi}(a|s)$ depends on both the states and the actions taken at those states, the effectiveness of $\epsilon$ depends on the state-action structure of the dataset. Below, we explain the intent behind our $\epsilon$ design choices based on the empirical observations from offline RL tasks. Figure 3(e) in Sec. 6.4 presents the final performance of FAC under different $\epsilon$ designs, the corresponding results are provided in the following table for convenience:
> |Tasks|Batch Adaptive|Dataset wide Constant|
> |-|-:|-:|
> |halfcheetah-medium-expert|101.9±5.6|71.9±4.8|
> |antmaze-medium-play|86.0±2.3|88.0±9.6|
> |antmaze-large-diverse|76.0±8.7|88.0±6.0|
>
> **1. Batch adaptive $\epsilon$:**
> Offline dataset often combine trajectories generated by behavior policies with different skill levels. To illustrate an extreme case, consider a dataset collected from medium-level and expert-level behavior policies that visit almost identical states. Although their state coverage is similar, the action distribution at the same state can differ significantly: the expert policy tends to exhibit substantially narrower action support than the medium policy.
>
> This creates a characteristic multi-modal, uneven state-action density: for the same state $s$, the medium-level actions occupy a relatively broad region, while expert-level actions form sharp modes with much higher density. If $\epsilon$ is fixed according to the lower-density component (e.g., medium behavior), the ratio $\hat{\beta}(a|s)/\epsilon$ can easily exceed 1 even for actions that are genuinely OOD around the expert mode. This causes the critic penalization weight to vanish for those OOD actions.
>
> We ran an experiment regarding $\epsilon$ on halfcheetah-medium-expert-v2, a dataset that, we hypothesize, exhibits narrow state coverage and highly uneven action densities. The result in the above table show that batch adaptive $\epsilon$ yields superior performance in this case. Our interpretation is that when the dataset contains multiple action modes per state with large density disparities, adapting $\epsilon$ to each mini-batch sample reflects the local density scale and prevents unintended disappearance of critic penalization.
>
> **2. Dataset wide constant $\epsilon$:**
> Conversely,  batch adaptive $\epsilon$ yields a bit degraded performance in tasks such antmaze-large-diverse, as seen in the above table. Our analysis is that Antmaze datasets differ fundamentally from halfcheetah-medium-expert. The D4RL antmaze dataset has very wide state coverage, however, at each state, the behavior policy generates only a small number of actions, often forming narrow and irregular local densities. This indicates that $\hat{\beta}_{\psi}(a|s)/\epsilon$ might fluctuate excessively under the batch adaptive $\epsilon$. The batch adaptive value of $\epsilon$ can become excessively large, which may penalize even actions that are essential for stitching trajectories from the start position to the goal position.
>
> To mitigate this effect, we introduced the dataset-wide constant $\epsilon$, providing a coherent lower bound on $\hat{\beta}_{\psi}(a|s)$, preventing over-penalization of states whose action samples are sparse and densities are uneven. With dataset-wide constant $\epsilon$, FAC achieves the highest performance on antmaze-large-diverse-v2, as seen in Figure 3(e) in Sec. 6.4. Furthermore, to avoid excessive hyperparameter tuning for $\epsilon$, we fixed the antmaze domain to use the dataset-wide constant $\epsilon$.
>
> **3. Robust performance across both $\epsilon$ schemes:**
> Outside of the above extremes, we observe that the two $\epsilon$ designs often perform similarly well. For instance, in antmaze-medium-play, both batch adaptive and dataset-wide constant $\epsilon$ achieve comparable performance even under the same critic penalization coefficient $\alpha$ and actor regularization coefficient $\lambda$. We conjecture that this result comes from the intermediate of behavior distribution of the two extreme datasets.

---

> ### Author Response · Authors · 2025-11-23
> **Authors' Response to Reviewer XLd9**
>
> ### R6. Offline-to-online evaluation (Question 4\&5)
>
> We evaluated FAC in the offline-to-online setting. This setting highlights how FAC behaves under distribution shift due to the limited coverage of the offline dataset. We trained FAC for 1M gradient steps and then fine-tuned it for additional 1M steps with online interaction. Experimental details for online finetuning are provided in Appendix G.3.3.
>
> We compare FAC with FQL, a strong offline-to-online baselines. The results are now presented in Figure 4 in Sec. 6.5 and also summarized in the table below (performance at 1M $\to$ one at 2M). Note that FAC reuses the hyperparameters as in the offline RL settings, whereas FQL uses the offline-to-online hyperparameters recommended in its original paper.
>
> On the OGBench and D4RL Antmaze tasks, FAC achieves strong performance in the offline-to-online setting. These results may come from our flow behavior proxy density based critic penalization design. The critic penalization design allows FAC to reduce or eliminate penalization for confident and ID-like actions, suggesting that FAC avoids excessively suppressing high Q-value neighborhoods near the dataset support boundary. This, in turn, may enable exploration in such high Q-value regions. In the D4RL Adroit tasks, however, FAC shows smaller fine-tuning improvement compared to FQL. We conjecture the reason as follows. Due to the notorious
> sparsity of the adroit datasets, FAC during the offline training phase provides only limited guidance about high Q-value regions, the critic penalization may unintentionally hinder exploration toward these promising neighborhoods.
>
> |Task Categories|FQL|FAC|
> |-|-:|-:|
> |OGBench (5 tasks)|32.9→72.3|61.1→84.5|
> |D4RL Antmaze (6 tasks)|81.9→96.3|90.1→99.1|
> |D4RL Adroit (4 tasks)|14.5→113.5|29.2→88.1|
>
>
> Thank you again for your time and effort.

---

### Official Review · Reviewer_K2us · 2025-11-01

**Soundness:** 3
**Presentation:** 3
**Contribution:** 3
**Rating:** 4
**Confidence:** 4

**Summary:**

This paper proposes FAC (Flow Actor-Critic), a new offline RL algorithm that integrates flow-based policies into the actor-critic framework. The motivation is that flow-based networks have much more expressibility than the Gaussian-based counterparts. The proposed method achieves state-of-the-art performance on the D4RL and OGBench benchmarks.

**Strengths:**

- The preliminary experiments strongly show that flow-matching can better estimate the density of the behavior policy.
- The proposed method is intuitive and seems to be a nice approach to incorporate flow-matching to offline RL.
- The proposed method shows strong performance not only on the D4RL benchmarks but also on the more recent OGBench benchmarks.

**Weaknesses:**

- Since the proposed method relies heavily on the accuracy of the density estimation from flow matching, there is some possibility that the proposed method may not scale well to higher-dimensional environments like pixel-based ones.
- Importantly, the proposed method introduces many new hyperparameters (alpha, lambda, clipped double Q-learning, and epsilon), and those hyperparameters are tuned for each task. This is not a fair comparison with the baselines. For example, CQL and IQL use the same hyperparameter per domain (Mujoco, Adroit, ...).

**Questions:**

- In the experiments of Figure 1, how does flow matching work when T=50?
- In Equation (8), there can be other designs for giving more weight to outlier actions, for example, exponentially increasing the weight for those actions. Could the authors explain why they chose this specific formulation?
- In L1261, the paper notes they chose the epsilon scheme according to the state coverage and the action diversity. Can the authors provide a clear definition of what the two terms mean?

---

> ### Author Response · Authors · 2025-11-23
> **Authors' Response to Reviewer K2us**
>
> We thank the reviewer for the insightful  comments, as well as the thoughtful suggestions that  reflects a deep and careful reading of the manuscript. Please note that the numbers of equations, figures, tables and sections   below refer to those in the revised paper available at openreview unless we give special remarks.
> Our responses to the comments are presented below.
>
> ### R1. Evaluation on pixel-based tasks (Weakness 1)
>
> We conducted experiments on pixel-based tasks and report the final performance evaluated at 500K gradient steps in Table 1 in Sec. 6. Experiments and Table 3 in Appendix F.1. As seen, our method yields superior performance to other algorithms even in the pixel-based tasks.  For this experiment, we use the same image processing components used in FQL (e.g., a small variants of IMPALA encoder [1]). All remaining hyperparameters are reused from the corresponding state-based tasks, in contrast to FQL, which uses visual-specific BC coefficient (refer to Table 6 in the FQL paper). For example, for visual-cube-single-play-singletask-task1-v0, we use the same hyperparameters as those used for cube-single-play-singletasks-v0. The performance of FAC and baselines is summarized in the table below.
> |Pixel-based Tasks|IQL|ReBRAC|FBRAC|IFQL|FQL|FAC(Ours)|
> |-|-:|-:|-:|-:|-:|-:|
> |visual-cube-single-play-singletask-task1-v0|70.0|**83.0**|55.0|49.0|81.0|82.0±5.2|
> |visual-cube-double-play-singletask-task1-v0|34.0|4.0|6.0|8.0|21.0|**48.0**±7.3|
> |visual-scene-play-singletask-task1-v0|97.0|98.0|46.0|86.0|98.0|**99.0**±2.0|
> |visual-puzzle-3x3-play-singletask-task1-v0|7.0|88.0|7.0|**100.0**|94.0| 95.0±2.0|
> |visual-puzzle-4x4-play-singletask-task1-v0|0.0|26.0|0.0|8.0|33.0|**56.0**±11.3|
> |**Average**|41.6|59.8|22.8|50.2|65.4|**76.0**±3.9|
>
> FAC achieves the highest overall performance across the pixel-based manipulation tasks with significant margin over other methods.  These results indicates that the performance gains achieved by FAC in state-based tasks also hold in the higher-dimensional pixel-based settings.
>
> [1] Espeholt, Lasse, et al. "Impala: Scalable distributed deep-rl with importance weighted actor-learner architectures." International conference on machine learning. PMLR, 2018.
>
> ### R2. The fidelity of flow matching model (Question 1)
>
> We investigated how the fidelity of the flow behavior proxy affects both sampling quality and density estimation.
> Figure 6 in Appendix D.1 shows the quality of generated samples and density estimation obtained from a single flow matching model with different step counts $T\in\{3,5,10,30,50\}$. Using the same trained flow matching model, we vary only the step counts of the Euler method for sampling and density estimation. When $T=\{3,5\}$, decreasing step counts leads to a clear degradation in both sampling and density estimation, with $T=3$ failing to reliably separate ID and OOD regions. In contrast, for $T\geq 10$, the sampling performance is satisfactory, and density estimates contrast is enough to distinguish ID and OOD regions.
>
> This observation is  because the Euler method servers as the numerical integration solver, and the step counts directly affects the fidelity of the approximated flow. When the learned flow matching model represents almost linear trajectories from base noises $x_0$ to data samples $x_1$,  step count 1 can be sufficient. However, for complex data distributions where the learned flow trajectories exhibit curve, higher curvature needs more integration steps to ensure sufficiently accurate sampling and density estimation.
>
> Based on this observation, we performed further experiment on  the step count of the Euler method in two RL tasks and evaluated the final performance of FAC,  shown in Figure 3(c) in Sec. 6.4. The results are summarized in the table below. In humanoidmaze-medium-navigate-task1, performance drops when $T<10$, whereas $T=30$ yields a slight improvement over $T=10$. Similarly, in puzzle-3x3-play-task4, performance steadily decreases as $T$ decreases below 10, whereas $T=30$ achieves the same 100\% success rate as $T=10$. These results demonstrate that $T=10$ provides a favorable trade-off between computational cost and performance.
> |Tasks|T=1|T=3|T=5|T=10|T=30|
> |-|-:|-:|-:|-:|-|
> |humanoidmaze-medium-navigate-task1|25.5±23.9|28.0±22.2|30.0±16.4|71.5±14.7|75.0±5.1|
> |puzzle-3x3-play-task4|86.5±21.6|91.5±18.4|96.5±9.9|100.0±0.0|100.0±0.0|

---

> ### Author Response · Authors · 2025-11-23
> **Authors' Response to Reviewer K2us**
>
> ### R3. Critic penalization weight (Question 2)
>
> Our key design principle in the weight (eq.8) is to use the ratio between the flow behavior proxy density $\hat{\beta}_{\psi}(a|s)$ of the action generated from the flow one-step actor and that of the dataset actions. As shown in Figure 1 in Sec. 3, the flow matching model can provide strong density contrast to distinguish ID and OOD regions. However, offline RL tasks might contain more diverse state-action pairs, making it difficult to rely on the absolute scale of the estimated densities. In practice, due to the limited dataset, the estimated density might be differ in scale from the true underlying densities on entire state-action space (e.g., being unnormalized). Correcting this mismatch would require estimating densities across the entire state-action space to compute an appropriate normalization constant, which seems infeasible.
>
> To avoid this issue, we design the weight using the ratio $\hat{\beta}_{\psi}(a|s)/\epsilon$, where  $\epsilon$ is defined by the densities of the dataset samples.
> This ratio design enables a gradually increase in critic penalization as actions become more OOD, while ratio above 1 indicate confident actions for which the penalization intentionally disappears.
>
> We agree on that various weight designs are possible, including exponential functions. Motivated by the reviewer's suggestion, we additionally designed two new weight functions. For clarity, we refer to the original eq.8 as the linear weight and the newly introduced weights as the convex and concave weights, respectively:
> - Linear (eq.8): $\max(0, 1-(\hat{\beta}(a|s)/\epsilon))$
> - convex: $\max(0, 1-(\log(\hat{\beta}(a|s)/\epsilon)^{\kappa}+1)/\log2)$, where $\kappa=1/2$
> - concave: $\max(0, 1-(\log(\hat{\beta}(a|s)/\epsilon)^{\kappa}+1)/\log2)$, where $\kappa=2$
>
> We note that three wights are linear or convex or concave on [0,1] of $\hat{\beta}_{\psi}/\epsilon$.
> The new result is available at
> Figure 3(d) in Sec. 6.4, comparing the three weight designs' performance of FAC. The results is also provided in the table below for convenience. As seen, the linear and concave weights achieve the same or similar performance, but the convex weight leads to performance degradation. This result indicates that the linear weight is a simple but effective choice.
> |Tasks|Linear(eq.8)|Convex|Concave|
> |-|-:|-:|-:|
> |humanoidmaze-medium-navigate-task1|71.5±14.7|59.5±20.1|73.5±8.8|
> |puzzle-3x3-play-task4|100.0±0.0|98.5±2.0|100.0±0.0|

---

> ### Author Response · Authors · 2025-11-23
> **Authors' Response to Reviewer K2us**
>
> ### R4. Dataset-driven method for $\epsilon$ (Question 3)
>
> The $\epsilon$ in the critic penalization weight $w^{\hat{\beta}}(s,a)$ determines whether a given state-action sample $(s,a)$ should be treated as reliably in-distribution. Since $\hat{\beta}_{\psi}(a|s)$ depends on both the states and the actions taken at those states, the effectiveness of $\epsilon$ depends on the state-action structure of the dataset. Below, we explain the intent behind our $\epsilon$ design choices based on the empirical observations from offline RL tasks. Figure 3(e) in Sec. 6.4 presents the final performance of FAC under different $\epsilon$ designs, the corresponding results are provided in the following table for convenience:
>
> |Tasks|Batch Adaptive|Dataset wide Constant|
> |-|-:|-:|
> |halfcheetah-medium-expert|101.9±5.6|71.9±4.8|
> |antmaze-medium-play|86.0±2.3|88.0±9.6|
> |antmaze-large-diverse|76.0±8.7|88.0±6.0|
>
> **1. Batch adaptive $\epsilon$:**
> Offline dataset often combine trajectories generated by behavior policies with different skill levels. To illustrate an extreme case, consider a dataset collected from medium-level and expert-level behavior policies that visit almost identical states. Although their state coverage is similar, the action distribution at the same state can differ significantly: the expert policy tends to exhibit substantially narrower action support than the medium policy.
>
> This creates a characteristic multi-modal, uneven state-action density: for the same state $s$, the medium-level actions occupy a relatively broad region, while expert-level actions form sharp modes with much higher density. If $\epsilon$ is fixed according to the lower-density component (e.g., medium behavior), the ratio $\hat{\beta}(a|s)/\epsilon$ can easily exceed 1 even for actions that are genuinely OOD around the expert mode. This causes the critic penalization weight to vanish for those OOD actions.
>
> We ran an experiment regarding $\epsilon$ on halfcheetah-medium-expert-v2, a dataset that, we hypothesize, exhibits narrow state coverage and highly uneven action densities. The result in the above table show that batch adaptive $\epsilon$ yields superior performance in this case. Our interpretation is that when the dataset contains multiple action modes per state with large density disparities, adapting $\epsilon$ to each mini-batch sample reflects the local density scale and prevents unintended disappearance of critic penalization.
>
> **2. Dataset wide constant $\epsilon$:**
> Conversely,  batch adaptive $\epsilon$ yields a bit degraded performance in tasks such antmaze-large-diverse, as seen in the above table. Our analysis is that Antmaze datasets differ fundamentally from halfcheetah-medium-expert. The D4RL antmaze dataset has very wide state coverage, however, at each state, the behavior policy generates only a small number of actions, often forming narrow and irregular local densities. This indicates that $\hat{\beta}_{\psi}(a|s)/\epsilon$ might fluctuate excessively under the batch adaptive $\epsilon$. The batch adaptive value of $\epsilon$ can become excessively large, which may penalize even actions that are essential for stitching trajectories from the start position to the goal position.
>
> To mitigate this effect, we introduced the dataset-wide constant $\epsilon$, providing a coherent lower bound on $\hat{\beta}_{\psi}(a|s)$, preventing over-penalization of states whose action samples are sparse and densities are uneven. With dataset-wide constant $\epsilon$, FAC achieves the highest performance on antmaze-large-diverse-v2, as seen in Figure 3(e) in Sec. 6. Furthermore, to avoid excessive hyperparameter tuning for $\epsilon$, we fixed the antmaze domain to use the dataset-wide constant $\epsilon$.
>
> **3. Robust performance across both $\epsilon$ schemes:**
> Outside of the above extremes, we observe that the two $\epsilon$ designs often perform similarly well. For instance, in antmaze-medium-play, both batch adaptive and dataset-wide constant $\epsilon$ achieve comparable performance even under the same critic penalization coefficient $\alpha$ and actor regularization coefficient $\lambda$. We conjecture that this result comes from the intermediate of behavior distribution of the two extreme datasets.

---

> ### Author Response · Authors · 2025-11-23
> **Authors' Response to Reviewer K2us**
>
> ### R5. Hyperparameters (Weakness 2)
>
> We would like to emphasize that the proposed method does not introduce substantially additional hyperparameter tuning. Below is the comparison with the primary baseline,  FQL, the closest flow-based offline RL method.
>
> For clipped double Q-learning, we follow exactly the same rule as FQL on the OGBench: "min" only for OGBench Antmaze, and "mean" for all remaining OGBench tasks. Furthermore, we use the standard "min" for all D4RL tasks, in contrast, FQL mixes the two ones across D4RL domains. our configuration is more consistent with the broader offline RL baselines.
>
> For the $\epsilon$, we use two designs but fix a single $\epsilon$ design per domain, avoiding excessive choice.
>
> The remaining continuous hyperparameters in FAC are the critic penalization coefficient $\alpha$ and actor regularization coefficient $\lambda$. We note that FQL also uses actor regularization coefficient $\lambda$. Moreover, as shown in Figure 3(b) in Sec. 6.4, FAC exhibits robust performance over a wide range of $\alpha$ and $\lambda$, indicating that these coefficients do not require fine-grained tuning.
>
> For OGBench tasks, the baselines, including IQL, perform task category-level tuning, as described in the FQL paper. For D4RL, relatively recent baselines (e.g. ReBRAC, EPQ, FQL, ..) also use task-specific hyperparameters. Even CQL uses several practical implementation choices (e.g. soft maximization of Q-values, auxiliary action distributions, and additional gradient computations).
>
> Taken together, FAC does not rely on more extensive hyperparameter tuning than the baselines. Thus, the comparison remains fair.
>
> Thank you again. We hope our response satisfies the reviewer's concerns.

---

### Public Comment · ~Jiayu_Xu4 · 2025-11-21

Nice work. I have a question: In Section 4.1, ϵ is important. Could you please provide a concrete example of the threshold values ​​on Ogbench and D4RL? Also, if ϵ is set to the minimum of beta_hat_density, then beta_hat_density / ϵ will always be greater than 1. Did I miss anything?

---

> ### Author Response · Authors · 2025-11-23
> **Authors' Response to Public Comment**
>
> We thank you for your interest in our work.
>
> The full hyperparmeters, including $\epsilon$, is provided in Table 6 and 7 in Appendix G.3.2.
>
> In the weight $w^{\hat{\beta}}(s,a)$, $\hat{\beta}(a|s)$ denotes the flow behavior proxy density of the actions generated by the flow one-step actor $\pi_{\theta}(\cdot|s)$, whereas $\epsilon$ is defined by the flow behavior proxy densities on the dataset samples $(s,a)\in\mathcal{D}$. If the flow actor generates actions that substantially deviate from those in the dataset, the resulting density can be lower than the minimum density over the dataset samples. In such cases, the ratio $\hat{\beta}_{\psi}(a|s)/\epsilon$ is less than 1, at which point critic penalization is activated.

---

### Author Response · Authors · 2025-11-23
**Common Response**

Dear reviewers

We sincerely thank all reviewers for their valuable comments. The feedback has greatly improved the manuscript and guided important revisions.

We have uploaded a revised version at Openreview, and all modified contents are highlighted in blue for ease of reference. Please note that the numbers of equations, figures, tables and sections below refer to those in the revised paper available at Openreview now.



The main revisions are summarized below.

- Pixel-based evaluation: offline RL evaluation results on the 5 pixel-based OGBench tasks have been added in Table 1 in Sec. 6.2 and Table 3 in Appendix F.1. We observe that our method yields superior performance to other algorithms even in the pixel-based tasks.

- Offline-to-online evaluation: the offline-to-online evaluation results across 15 tasks has been added in Figure 4 in Sec. 6.5 and Figure 7 in Appendix F.2. We observe that FAC shows strong offline-to-online performance.


- Ablation on the fidelity of flow behavior proxy: to analyze the influence of numerical accuracy of the flow behavior proxy, we added:
(i) data sampling and density estimation quality of the flow matching model under the Euler method with different step counts in Figure 6 in Appendix D.1, and
(ii) the influence of flow proxy fidelity on the performance of FAC in Figure 3(c) in Sec. 6.4.

- Ablation on critic penalization weights: the performance of FAC across critic penalization weights is added in Figure 3(d) in Sec. 6.4.

- Ablation on dataset-driven method for $\epsilon$ schemes: the performance of FAC under different $\epsilon$ designs is added in Figure 3(e) in Sec. 6.4.

- Computational costs: a comparison of training time, GPU memory usage, and inference time of FAC and baselines is  provided in Figure 8 in Appendix H.

- Comparison with baselines with sequence model architectures: a comparison of FAC against Transformer-based and state space model-based baselines in Table 8 in Appendix I.

- Theoretical analysis: a theoretical analysis of the FAC actor, critic, operator in Proposition 2 and Theorem 1\&2 in Appendix J.

- Comparison of baselines and FAC in a continuous action bandit problem: a comparison of FAC and baselines in terms of Q-estimates and induced policies in Appendix C.

We believe that our work indeed sets a state-of-the-art benchmark in the area of offline reinforcement learning, and hope our response together with the revised paper satisfies all the concerns of the reviewers. Thank you.

---

### Author Response · Authors · 2025-12-03
**Author Summary for Area Chair**

Dear Area Chair,

We thank you for handling our submission.

In this paper, we proposed Flow Actor-Critic (FAC), a novel actor-critic method for offline RL based on flow model. Our method leverages an expressive flow model not only for the actor but also crucially for the acquisition of a conservative critic, a key component in mitigating value overestimation. We introduced a new critic penalization loss based on the behavior density of flow model, where a critic penalization weight adaptively suppresses Q-values only when the behavior density falls below a dataset-driven threshold. This thresholding method provides a criterion governing when penalization is applied. Our approach demonstrates strong performance across diverse datasets in offline RL benchmarks, and sets the state-of-the-art performance in OGBench with large margins over existing methods.

We would like to briefly summarize the major reviewers' comments and our response and revision below.

**Theoretical analysis and visual validation of the accuracy of the proposed method**

Reviewer FJQh requested a more rigorous theoretical analysis of the proposed critic penalization loss and suggested adding clear visual evidence of improved Q function estimation. We included a theoretical analysis of the critic loss in Appendix J and added visual comparisons of our method and baseline algorithms in Appendix C of the revised paper, showing the superiority of our method.



**The fidelity of flow behavior proxy**

Reviewer K2us questioned how the fidelity of the flow matching model depends on the step counts of Euler method used in the numerical ODE solver. We provided results in Figure 6 and detailed analysis in our response. Reviewer XLd9 and Ypn5 asked how potential inaccuracies in the flow model influence the performance of our method. To address this, we conducted ablation studies presented in Figure 3(c).



**Critic penalization weight**

Reviewer K2us asked for an explanation about the specific design choice of the proposed weight. In our response, we explained the rationale behind the weight and detailed the key design principle. We further provided ablation studies on alternative weight formulations, with the results presented in Figure 3(d).



**Dataset-driven threshold scheme**

Reviewer K2us asked for a rationale behind our dataset-driven threshold choice, and Reviewer XLd9 and Ypn5 questioned how this threshold scheme affects the performance of our method. To address these points, we provided empirical results for different threshold choices in Figure 3(e), and, based on these results, we offered an insightful rationale and detailed explanation of the proposed dataset-driven threshold designs in our response.



**Additional Experiments**

Reviewer K2us raised concerns about the applicability of our method to pixel-based datasets. We provided evaluation results on 5 pixel-based OGBench tasks in Table 1 and 3. Reviewer XLd9 questioned about the feasibility of our method in online finetuning setting. We conducted offline-to-online experiments and presented the results in Figures 4 and 7. It is observed that our method yields superior performance even in these tasks.


We believe these clarifications, analyses, and additional experiments adequately address the reviewers' concerns and further strengthen our work.

We sincerely appreciate the constructive feedback of the reviewers and thank the Area Chair for your dedications.

Sincerely,

Authors

---

### Meta-Review · Area_Chair_QV71 · 2026-01-06

**Summary:**

This paper proposes Flow Actor-Critic (FAC), an offline RL method that uses a flow-based behavior proxy not only for expressive action generation but also for density estimation to identify OOD regions and apply density-aware critic penalization while regularizing the actor toward the behavior support. Reviewers broadly found the idea clean and results strong across D4RL and OGBench, but the key concerns were fairness/hyperparameter tuning (task-specific tuning vs baselines), scalability and reliability of density estimation (including pixel-based/high-dimensional settings), missing compute cost discussion, and the need for clearer theoretical justification and stronger evidence that FAC improves Q estimation under OOD rather than being an incremental extension of prior flow methods (e.g., FQL).

**Reviewer Concerns:**

The rebuttal substantially addressed major issues: it added pixel-based OGBench experiments showing strong performance with reused hyperparameters, provided ablations on Euler step count and penalization weight designs, clarified and studied the threshold/epsilon scheme (batch-adaptive vs dataset-wide), and argued hyperparameter burden is comparable to recent baselines. For theory and Q-quality evidence, the authors added a more rigorous analysis (fixed point / support-constrained policy claims) and replaced/expanded Q-function illustrations with a bandit setting where ground-truth Q is known; they also added computational comparisons (time/memory/inference) versus diffusion and other baselines, plus added comparisons to sequence-model baselines (DT/QDT/Reinformer/Decision Mamba) in appendix. Remaining concerns are mainly about positioning as incremental over FQL for some reviewers and whether the Q-accuracy claim is fully general beyond the provided theory/diagnostic examples, but the rebuttal meaningfully reduced the earlier evidence gaps.

**Reviewer Scores:**

K2us likely 4→6 (pixel-based results, ablations, and clearer epsilon/weight design address core weaknesses and fairness concerns). XLd9 likely 6→7 (compute costs and density-fidelity evidence added; novelty concern partially alleviated). Ypn5 likely stays 8→8 (already strong accept; minor threshold study addressed). FJQh likely 6→6 or 7 (they raised continued concerns, but authors added theory + bandit Q-ground-truth evidence + sequence-model comparisons; conservative estimate is small or no change).

---

### Decision · Program_Chairs · 2026-01-26

Accept (Poster)